

# Ensemble numerical simulation of permafrost over the Tibetan Plateau from Flexible Permafrost Model: 1950–2023

Wen Sun and Bin Cao

State Key Laboratory of Tibetan Plateau Earth System Environment and Resources (TPESER), National Tibetan Plateau Data Center (TPDC), Institute of Tibetan Plateau Research, Chinese Academy of Sciences, Beijing, China

**Correspondence:** Bin Cao (bin.cao@itpcas.ac.cn)

**Abstract.** Permafrost remains a largely subsurface phenomenon, and understanding its dynamics as well as influences under a warming climate heavily relies on numerical simulations. However, this task presents significant challenges as the state-of-the-art land surface models are weak in their ability to represent permafrost processes. In this study, we introduce a new land surface scheme specifically designed for permafrost applications, the Flexible Permafrost Model (FPM). This model serves as

an adaptable framework for implementing innovative parameterizations of permafrost-related physics. The FPM accounts for both vertical and lateral heat flow at and below the soil surface, while simultaneously resolving the land-atmosphere energy exchanges through comprehensive treatment of radiative balance and turbulent flux dynamics. We simulate the ground thermal regime and test the model with a network of permafrost measurements across the Tibetan Plateau.

Our result yields root mean square error values of 1.1 m for the thickness of the active layer and 1.5 °C for the mean annual
ground temperature of permafrost. We estimate that the current extent of permafrost (2010–2023) on the Tibetan Plateau is approximately $1.15 \pm 0.02 \times 10^6$ km$^2$, which aligns closely with published estimates. Long-term simulations indicate that the permafrost temperature has increased by 0.26 °C since 1980 with a decreased area of $13.2 \times 10^4$ km$^2$ ($\sim$10.5 %). These ensemble simulations provide valuable information on the dynamics of permafrost over the Tibetan Plateau. Furthermore, our findings suggest that current land surface models, which utilize shallow soil columns, are insufficient for permafrost simulations
over the Tibetan Plateau due to the typically deep active layer (that is, $2.88 \pm 0.95$ m by mean) and may not be suitable for future projections.

## 1 Introduction

Permafrost regions occupy more than one-fifth of the exposed land area in the Northern Hemisphere (Gruber, 2012). Long-term records revealed steady warming of permafrost over the past several decades at a global scale (Biskaborn et al., 2019). This
has led to significant degradation of the permafrost, such as a deepening active layer (Cao et al., 2018; Biskaborn et al., 2019), decreased permafrost area (Chadburn et al., 2017; Sun et al., 2022; Langer et al., 2024), expanded thermokarst (Wellman et al., 2013; Chen et al., 2021), and potential carbon decomposition (Schuur et al., 2015). Permafrost degradation has considerable influences on ecosystems, hydrological systems, and the integrity of the infrastructure (Hjort et al., 2018; Walvoord et al.,



2019; Schuur and Mack, 2018). Therefore, detailed investigations of changes in permafrost in response to a warming climate
are crucial for sustainable management and adaptation strategies.

Despite permafrost's importance, direct permafrost measurements, such as borehole temperature, are rare due to harsh environments and high costs (Biskaborn et al., 2015). This is especially true on the Tibetan Plateau (TP), where complex terrain and high altitudes impose further constraints on permafrost research (Cao et al., 2017b, 2019b). For these reasons, there is often a lack of essential *in situ* information to develop suitable statistical models (Zhao et al., 2021; Cao et al., 2021). Therefore,
process-based simulation is an increasingly important tool for transient assessment of permafrost conditions and dynamics.

The TP, also known as the Third Pole, has the largest extent of permafrost in the low-middle latitudes (Cao et al., 2019b). Significant efforts have been made to understand the permafrost changes over the TP based on numerical simulations. The process-based models used for recent transient permafrost simulation over the TP can be generally divided into geothermal numerical models (i.e., GIPL model) and the common land surface models (i.e., CLM and Noah-MP). The geothermal nu-
merical models typically have rich permafrost-specific processes, such as suitable numerical solver in heat transfer with soil phase changes (Lan et al., 2025), deep soil column (tens to hundreds of meters), and well-defined lower boundary, but lack representation of land-atmosphere interactions (i.e., Qin et al., 2017; Sun et al., 2023). On the other hand, the land surface models benefits from the consideration of land-atmosphere processes, and therefore outperform in describing the responses and influences of permafrost to climate warming (i.e., Guo et al., 2018; Wu et al., 2018; Zhang et al., 2021; Cao et al., 2022).
Recently, a few permafrost-specific land surface scheme models–combining the advantages of these two types of models–were proposed. The stand-alone models yield promising potential for application to cross-scale permafrost processes (Fiddes et al., 2015; Westermann et al., 2016; Langer et al., 2024). However, such stand-alone permafrost simulations are rare over the TP, and are restricted to site scale and short coverage (i.e., Pan et al., 2016; Zheng et al., 2020).

In this study, we introduce a new land surface scheme specifically designed for permafrost applications, the Flexible Per-
mafrost Model (FPM). This model serves as a flexible platform to explore novel parameterizations for a variety of permafrost processes. As proof of the suitability of the new model, we perform an initial evaluation of its performance in the context of reproduction of the long-term (1950–2023) permafrost thermal regime over the TP. Specially, this study

1. gives a detailed description of the model conceptualization, structure, and parameterization;

2. evaluates the model performance in reproducing permafrost characteristics based on the ensemble approach, such as
active layer thickness (ALT), and the thermal state;

3. interprets current conditions and historical changes of permafrost in response to climate change from the stand-alone simulations;

4. proposes insights for future model developments.



## 2  Flexible Permafrost Model (FPM)

The FPM is land-surface scheme designed, so that algorithms and process parameterizations can easily be transferred, and the ensemble simulation could be produced based on specific scientific objectives. FPM accounts for both vertical and lateral heat flow with phase change at and below the soil surface, while also describing the energy exchange with the atmosphere by considering radiative and turbulent fluxes. The application of FPM with lateral heat is provided in Sun et al. (2023). In this study, we give the detailed introduction of FPM with 1D heat flow and demonstrate its suitability for large-scale permafrost studies.

### 2.1  Surface energy balance

A physically-based surface energy balance scheme for different land surface cover types with varying snow regimes and properties was coupled to FPM, and was formulated as:

$$(1-\alpha)Q_{si} + Q_{li} + Q_{le} + Q_h + Q_e + Q_c = Q_m \tag{1}$$

where $Q_{si}$ is the incoming shortwave radiation, $Q_{li}$ is the incoming longwave radiation, $Q_{le}$ is the emitted longwave radiation, $Q_h$ is the turbulent exchange of sensible heat, $Q_e$ is the turbulent exchange of latent heat, $Q_c$ is the energy transport due to conduction, and $Q_m$ is the energy flux available for melt. All the above energy terms have units of $W\,m^{-2}$. The $\alpha$ $(-)$ is the surface albedo, obtained as a fraction-weighted average of albedo from snow-free ($\alpha_g$) and snow-covered ($\alpha_{sn}$) areas.

$$\alpha = (1 - SCF) \cdot \alpha_g + SCF \cdot \alpha_{sn} \tag{2}$$

where SCF $(-)$ is the snow cover fraction, and $\alpha_g$ is from MODIS products whenever snow is not present (Table 1).

The $Q_{si}$ and $Q_{li}$ are either from observations or from a reanalysis dataset (Table 1). The $Q_{le}$ is computed under the assumption that ground (snow) emits as a gray body:

$$Q_{le} = -\varepsilon_s \sigma T_{s0}{}^4 \tag{3}$$

where $\varepsilon_s$ $(-)$ is the surface emissivity, $\sigma$ is the Stefan-Boltzmann constant of $5.67 \times 10^{-8}\ W\,m^{-2}\,K^{-4}$, and $T_{s0}$ (K) is ground or snow surface temperature. The $\varepsilon_s$ was set to 0.92 for ground surface, and 0.98 for snow surface (Fleagle and Businger, 1981).

The turbulent exchange of sensible heat $Q_h$ is given by Price and Dunne (1976):

$$Q_h = \rho_a c_p D_h (T_a - T_{s0}) \tag{4}$$

where $\rho_a = 1.225\ kg\,m^{-3}$ is the air density, $c_p = 1004\ J\,kg^{-1}\,K^{-1}$ is the specific heat of air (Fleagle and Businger, 1981), and $T_a$ (K) is near-surface air temperature. The exchange coefficients for heat $D_h$ $(-)$ can be estimated as:

$$D_h = \frac{\kappa^2 u_z}{\left(\ln\left(z/z_0\right)\right)^2} \tag{5}$$





where $\kappa$ is the Von Karman's constant with a value of 0.4 (Fleagle and Businger, 1981), $u_z$ (m) is the wind speed at the instrument height z (m), and $z_0$ (−) is the roughness length as 0.015 m for ground surface and 0.001 m for snow surface following Ling and Zhang (2004).

The latent heat flux was calculated via the Priestley-Taylor method (Priestley and Taylor, 1972).

$$Q_e = S \cdot \alpha_{pt} \frac{\Delta(Q_n - Q_c)}{\Delta + \gamma} \tag{6}$$

where S (−) is the evaporation stress factor, $\alpha_{pt}$ (−) is the Priestly–Taylor coefficient, $Q_n$ (W m$^{-2}$) is the net radiation, $\Delta$ (Pa K$^{-1}$) is the slope of the saturation vapor pressure-temperature curve, and $\gamma$ (Pa K$^{-1}$) is psychrometric constant. $\Delta$ is determined following Dingman (2015):

$$\Delta = \frac{4098 e_s}{(T_a - T_{frz} + 237.3)^2} \tag{7}$$

where $T_{frz} = 273.15$ K is the freezing point temperature, and $e_s$ (Pa) is saturation vapor pressure derived following Dingman (2015):

$$e_s = 611 \exp\left(\frac{17.3\,(T_a - T_{frz})}{T_a - T_{frz} + 237.3}\right) \tag{8}$$

the $\gamma$ is calculated using the formula proposed by Brunt (2011):

$$\gamma = \frac{c_p P}{\varepsilon L_v} \tag{9}$$

where P (Pa) is atmospheric pressure, $L_v = 2.471 \times 10^6$ J kg$^{-1}$ is the latent heat of vaporization, and $\varepsilon$ is a constant of 0.622 (Dingman, 2015). For each grid, the latent heat flux is treated separately for the bare ground surface, vegetation, and snow. The detailed parameterizations are in Appendix A.

Heat conduction through the snow layer or ground surface ($Q_c$) was given as Liston and Hall (1995):

$$Q_c = -(T_{s0} - T_g)\left(\frac{z_{sn}}{k_{sn}} + \frac{z_g}{k_g}\right)^{-1} \tag{10}$$

where $T_g$ (K) is the ground temperature at the $z_g$ (0.1 m) depth, and $z_{sn}$ is the thicknesses of snow (if present). The $k_{sn}$ and $k_g$ (W m$^{-1}$ K$^{-1}$) are the thermal conductivity of the snow and ground at the depth of $z_g$, respectively.

The surface energy balance and heat conduction constitute a coupled non-linear equation system, and was solved iteratively for the surface temperature $T_{s0}$, using the Newton-Raphson method.

## 2.2 Snow scheme

To represent the influences of seasonal snow on soil thermal regime, a bulk snow scheme with static snow density was introduced. The snow layer was discretized to multiple layers with a vertical resolution of 0.01 m for heat transfer. The snow albedo is treated separately for non-melting and melting conditions following Douville et al. (1995). For non-melting conditions, a





linear decrease is assumed for $\alpha_{sn}$, while an exponential decrease is assumed for melting snow due to the presence of liquid water.

$$\alpha_{sn}^{t+\Delta t} = \begin{cases} \alpha_{sn}^t - 0.008\Delta t, & non-melting \\ \alpha_{sn}^{min} + (\alpha_{sn}^t - \alpha_{sn}^{min})e^{-0.24\Delta t}, & melting \\ \alpha_{sn}^{max}, & \Delta snd \geq 0.01 \end{cases} \tag{11}$$

where $\alpha_{\text{sn}}^{\max} = 0.85$ denotes the maximum snow albedo or the fresh snow albedo, while $\alpha_{\text{sn}}^{\min} = 0.50$ is the minimum snow albedo for old snow, and $\Delta$snd (m) refers to snow depth difference for simulation step $\Delta$t (i.e., 1 day). If there is a significant snowfall, i.e., $\Delta$snd $\geq 0.01$ m, snow albedo was reset to the maximum by assuming the surface is completely overlaid by the fresh snow.

The snow cover fraction was given as

$$SCF = min\left(\frac{snd}{snd_{cr}}, 1\right) \tag{12}$$

where snd (m) is the snow depth, and $\text{snd}_{\text{cr}} = 0.1$ m is the minimum snow depth that ensures complete coverage of the grid cell.

The snow volumetric heat capacity ($\text{CV}_{\text{sn}}$, $\text{MJ}\,\text{m}^{-3}\,\text{K}^{-1}$) and thermal conductivity are treated as functions of snow density ($\rho_{\text{sn}}$, $\text{kg}\,\text{m}^{-3}$) following Douville et al. (1995):

$$CV_{sn} = CV_i \frac{\rho_{sn}}{\rho_i} \tag{13}$$

$$k_{sn} = k_i \left(\frac{\rho_{sn}}{\rho_w}\right)^{1.88} \tag{14}$$

where $\text{CV}_{\text{i}} = 1.93 \times 10^6$ ($\text{J}\,\text{m}^{-3}\,\text{K}^{-1}$) is volumetric heat capacity for ice, $\text{k}_{\text{i}} = 2.22$ ($\text{W}\,\text{m}^{-1}\,\text{K}^{-1}$) is the thermal conductivity of ice, $\rho_{\text{i}} = 920\ \text{kg}\,\text{m}^{-3}$ is ice density, and $\rho_{\text{w}} = 1000\ \text{kg}\,\text{m}^{-3}$ is water density (Campbell and Norman, 2000). The current model does not consider the snow densification, and the static snow density of $250\ \text{kg}\,\text{m}^{-3}$ from *in situ* measurement was used (Wang et al., 2024). Snow ablation was heavily simplified by directly using the snow depth from reanalysis. For the snow-melt season, if the snow temperature was found positive after conducting heat conduction, it was set to 273.15 K.

## 2.3 Ground heat conduction and phase change

The ground temperature T (K) changes over time (t) and depth (z, m), and is numerically solved by the heat conduction for energy transfer and phase change determined by Fourier's law:

$$C\frac{\partial T}{\partial t} = \frac{\partial}{\partial z}\left(k\frac{\partial T}{\partial z}\right) \tag{15}$$

The latent heat during phase change is taken into account through an apparent volumetric heat capacity:

$$C = CV_s + L\frac{\partial \theta_u}{\partial T} \tag{16}$$





where C and $CV_s$ are the apparent volumetric heat capacity and volumetric heat capacity of soil ($\mathrm{J\,m^{-3}\,K^{-1}}$), respectively, L ($\mathrm{J\,m^{-3}}$) is the volumetric latent heat of fusion for ice, and $\theta_u$ ($\mathrm{m^3\,m^{-3}}$) is the volumetric unfrozen water content or super-cooled water. The unfrozen water was parameterized following Niu and Yang (2006):

$$\theta_u = \theta_{sat}\left\{\frac{10^3 L_f\left(T - T_{frz}\right)}{gT\psi_{sat}}\right\}^{-\frac{1}{b}} \tag{17}$$

where $\theta_{sat}$ ($\mathrm{m^3\,m^{-3}}$) is the saturated soil moisture, $L_f = 0.334 \times 10^6\ \mathrm{J\,kg^{-1}}$ is mass specific latent heat of water, $g = 9.80665$ ($\mathrm{m\,s^{-2}}$) is the acceleration due to gravity, $\psi_{sat}$ (mm) is the saturated soil matric potential depending on the soil material properties, and b is the Clapp-Hornberger parameter (Appendix B). FPM implements the nonlinear heat-transfer equations in Cartesian coordinates.

Thermal properties of the soil are assumed to be a weighted combination of different components of the soil column. The
145 $CV_s$ is calculated from the volumetric fractions of the constituents as follows Westermann et al. (2016):

$$CV_s = \sum_n f_n CV_n \tag{18}$$

where subscripts n = m, o, w, i, a, and g refer to soil constituents of mineral, organic, water, ice, air, and gravel. In this context, $f_n$ ($\mathrm{m^3\,m^{-3}}$) and $C_n$ ($\mathrm{J\,m^{-3}\,K^{-1}}$) represent the volumetric contents and the volumetric heat capacities for each component (Table E1). Similarly, the thermal conductivity of the soil $k_s$ ($\mathrm{W\,m^{-1}K^{-1}}$) is calculated based on their composition following
Cosenza et al. (2003):

$$k_s = \left(\sum_n f_n \sqrt{k_n}\right)^2 \tag{19}$$

where $k_n$ is the thermal conductivity ($\mathrm{W\,m^{-1}K^{-1}}$) for each soil component (Appendix E).

## 3 Model setting up and ensemble simulation

### 3.1 Soil profile

The soil column is discretized to 172 layers with a total depth of 150 m. The soil vertical grid size increased from 0.01 m for subsurface to 5.0 m for deep soil (Table D1). We use the soil texture of the Global Soil Dataset for Earth System Models (GSDE) from Dai et al. (2019) as it additionally provides the soil gravel content, which is prevalent over the TP. GSDE has 8 soil layers with a total depth of 3.8 m. The information of the last layer was extended to deeper soil, but was further refined based on the bedrock dataset from Shangguan et al. (2017). Note that, we assume soil organic matter is absent for soil below 3
160 m as is normally done in Earth system models (c.f., Chadburn et al., 2015).

### 3.2 Soil water content

For the current version, the static soil moisture is used. To specify the vertical water distribution within the soil column, we used sub-grid parameterizations from SURFEX and CryoGridLite (Decharme et al., 2006; Masson et al., 2013; Langer et al.,





2024). Here, we distinguished four hydrological layers in the subsurface, from the uppermost surface to deep layer, including the: 1) root zone; 2) vadose layer, extending from the root layer downward to the lower boundary or saturated table/bedrock (if present); 3) saturated layer, which lies between the depth of groundwater table and bedrock; and 4) bedrock layer, extends down to the lower boundary. Note that the presence and depth of the saturated layer is estimated based on the groundwater table information from Fan et al. (2013). The root depth is set between 0.05 m for bare soil and 0.5 m for high vegetation, and the typical depth for different vegetation cover is from GEOtop (Endrizzi et al., 2014).

In the root layer, the water content $\theta_R$ ($\mathrm{m^3\,m^{-3}}$) is estimated as the ensemble mean of five remote sensing-based products (Table 1, details see Sec. 3.3). The water content for the vadose layer $\theta_v$ ($\mathrm{m^3\,m^{-3}}$) is determined based on field capacity $\theta_{fc}$ ($\mathrm{m^3\,m^{-3}}$) and soil porosity $\phi$ ($\mathrm{m^3\,m^{-3}}$), and an ensemble range is used (see Sec.3.3). Please see Appendix B for the parameterizations of soil properties. In the saturated layer, the water content ($\theta_{sat}$) is equal to $\phi$. The water content of 0.05 $\mathrm{m^3\,m^{-3}}$ was used for the bedrock (Gubler et al., 2013).

## 3.3 Ensemble simulations

The ensemble simulations–allowing degrees of parameter uncertainties–are widely used for Earth system studies as they generally outperform individual simulations (Cao et al., 2019a; Langer et al., 2024). In this study, the ensemble simulation is produced using reasonable ranges of parameters (Table 1, Table 2). Since FPM currently does not consider the soil water balance, and because of the low confidence in the available datasets and parameterizations, the ensemble ranges are used for the related model parameters. Both the ensemble range of soil moisture in root layer and vadose zone are assumed here to allow possible uncertainties. The $\theta_R$ is derived as an ensemble mean of five state-of-the-art products (Table 1), and the ensemble was produced by $\pm$ standard deviation of all these products. The baseline of $\theta_v$ was determined as the mean of $\theta_{fc}$ and $\theta_{sat}$ in previous studies (e.g., Langer et al., 2024). This means the soil moisture in the vadose layer is in the middle of $\theta_{fc}$ and $\theta_{sat}$, coincidentally. We followed this algorithm, but allowed the propagation of uncertainty into model results. This is qualitatively described by a dry and wet variants of the $\theta_v$ parameter used (Table 2). We emphasize that the ensemble was chosen after considerable tests and comparisons but ultimately remains a subjective choice at this time. The range of the above selected parameters are used for the 45-member ensemble simulation to represent a wide range of permafrost conditions.

## 3.4 Model settings

The simulation was conducted at a time step of 1 day. A lower boundary condition of geothermal heat flux from Davies (2013) is used (Table 1). The first decade (July 1950–June 1960) climate forcing was used to spin up the model by running it 100 times (1000 years). Simulation performance was measured by the mean bias (BIAS) as well as root mean square error (RMSE), and more details can be found in Appendix C.



**Table 1.** Climate forcing and input datasets used in Flexible Permafrost Model (FPM).

| Input parameter | Dataset | Period | Resolution | Source/reference |
|---|---|---|---|---|
| Climate forcing | ERA5-Land[1] | 1950–2023 | 0.10° | Muñoz-Sabater et al. (2021) |
| *Surface cover* | | | | |
| Vegetation optical depth | VOD Climate archive | 2002–2018 | 0.25° | Moesinger et al. (2020) |
| Leaf area index | Reprocessed MODIS version 6.1 LAI | 2000–2021 | 0.05° | Myneni et al. (2021) |
| Vegetation type | 1:1 million vegetation map of China | static | 1/120° | Hou (2019) |
| Surface albedo | Global surface blue-sky albedo | 2001–2020 | 0.05° | Jia et al. (2022) |
| *Soil profile and moisture* | | | | |
| Surface soil moisture | ASCAT | 2007–2022 | 0.11° | SAF (2020) |
| | AMSR2–LPRM | 2012–2023 | 0.25° | Parinussa et al. (2015) |
| | ESA CCI SM | 2000–2022[2] | 0.25° | Dorigo et al. (2017) |
| | SMOC–IC | 2010–2021 | 0.25° | Fernandez-Moran et al. (2017) |
| | SMAP–L3 | 2015–2023 | 0.37°×0.44° | O'Neill et al. (2021) |
| Soil texture | Global Soil Dataset for Earth System Models | static | 1/120° | Dai et al. (2019) |
| Bedrock depth | Global depth to bedrock | static | 1/120° | Shangguan et al. (2017) |
| Watertable depth | Groundwater table depth | static | 1/120° | Fan et al. (2013) |
| Geothermal heat flux | Global Map of Solid Earth Surface Heat Flow | static | 2° | Davies (2013) |

[1] Correction is applied to the snow depth from ERA5-Land (see Sec.4.2).

[2] To be consistent with the other soil moisture products, only the period of 2000–2022 are used here, although ESA CCI provides a longer period.

**Table 2.** Soil moisture ($\mathrm{m}^3\,\mathrm{m}^{-3}$) parameters selected for ensemble simulations. The minimum and the maximum indicate the parameter ensemble range, and base indicates the standard choice used in model simulation.

| Soil layer | Root layer | Vadose layer |
|---|---|---|
| Symbol | $\theta_R$ | $\theta_v$ |
| Base | ensemble mean[1] | $\frac{\theta_{\mathrm{sat}}+\theta_{\mathrm{fc}}}{2}$ |
| Minimum | $-$std.[2] | $-0.1(\theta_{sat}-\theta_{fc})$ |
| Maximum | $+$std. | $+0.1(\theta_{\mathrm{sat}}-\theta_{\mathrm{fc}})$ |
| Step | $\frac{\mathrm{std.}}{4}$ | $0.05(\theta_{\mathrm{sat}}-\theta_{\mathrm{fc}})$ |

The footnote of [1] and [2] mean the ensemble mean and standard

deviation (std.) of five remote-sensing-based soil moisture in Table 1.



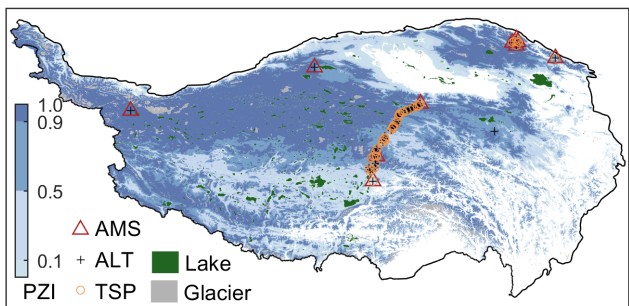

**Figure 1.** *In situ* observations used for model evaluation, including comprehensive observation sites with both the active layer soil temperature and atmospheric observations from the automatic meteorological system (AMS), active layer thickness (ALT) from soil temperature profile, and the permafrost thermal state (TSP) from boreholes. The permafrost zonation index (PZI) is from Cao et al. (2019b).

## 3.5 Diagnosis and analyses of permafrost characteristics

The permafrost is diagnosed following the definition of permafrost, i.e., the daily soil temperature of a simulation pixel remains at or below 0 °C for two or more years at given depth. Instead of directly using presence/absence of permafrost for individual ensemble member, the permafrost zonation index (PZI), as a fraction (0–1) corresponds to the 45 simulation members identified the probability of permafrost presence for each grid, is introduced to represent permafrost extent here. In such, the PZI can be used to quantitatively explore permafrost changes for each grid. More details of permafrost probability estimate could be found from Obu et al. (2019); Burke et al. (2020). In this study, we especially focus on the thermal state of permafrost at a depth of 3 m as the near-surface permafrost treated in most land surface models (Burke et al., 2020), and 15 m as the permafrost mean annual ground temperature (MAGT). The permafrost extent for above 100 m is additionally used as reference. The glaciers and lakes from ERA5-Land are masked before permafrost extent analyses. The ALT is estimated from linear interpolation of daily soil temperature. In this study, the period of 2010–2023 is used as the current condition for permafrost.

## 4 Data

### 4.1 Climate forcing

FPM is driven by climate forcing from reanalysis, including: near-surface air temperature, wind speed, incoming shortwave radiation, incoming longwave radiation, and atmospheric pressure. Snow depth is also required as FPM currently does not consider the snow mass balance. In FPM, the historical climate is taken from the ERA5-Land datasets, produced by the European Centre for Medium-Range Weather Forecasts (ECMWF, Table 1). ERA5-Land is a enhanced land component of ERA5 with a spatial resolution of 0.1° and a coverage from 1950 to the present (Muñoz-Sabater et al., 2021). The reanalyses were evaluated against the in situ observations (Appendix D).



**Table 3.** Metadata for the synthesis sites with both permafrost and atmospheric forcing measurements, including latitude (Lat, ° N), longitude (Lon, ° E), elevation (Ele, m), vegetation type, mean soil moisture (SM, $\mathrm{m^3\,m^{-3}}$) in growth season, mean annual ground temperature (MAGT, °C), active layer thickness (ALT, m), and measurement period.

| Site | Lat | Lon | Ele | Vegetation | MAAT | SM | MAGT | ALT | Period | References |
|------|-----|-----|-----|------------|------|----|------|-----|--------|-----------|
| AYK | 37.54 | 88.80 | 4300 | Alpine dessert | −5.2 | 0.05 | −1.70 | / | 2014–2018 | |
| LDH | 31.82 | 91.74 | 4808 | Alpine swamp meadow | −2.3 | 0.41 | / | 1.2 | 2002–2018 | |
| TGL | 33.07 | 91.94 | 5100 | Alpine meadow | −4.7 | 0.14 | −1.15 | 3.3 | 2006–2013 | Zhao et al. (2021) |
| TSH | 35.36 | 79.55 | 4740 | Alpine dessert | −6.0 | 0.09 | −2.63 | 1.0 | 2015–2019 | |
| XDT | 35.72 | 94.13 | 4538 | Alpine meadow | −3.6 | 0.32 | −0.54 | 1.4 | 2011–2018 | |
| EboA | 38.00 | 100.92 | 3691 | Alpine swamp meadow | −2.6 | 0.66 | −0.68 | 0.8 | 2011–2021 | |
| PT1 | 38.78 | 98.75 | 4128 | Alpine swamp meadow | −7.4 | 0.40 | −1.76 | 1.6 | 2011–2021 | |
| PT5 | 38.81 | 99.03 | 3691 | Alpine meadow | −2.3 | 0.33 | 0.03 | 3.6 | 2011–2021 | Cao et al. (2018) |
| PT6 | 38.95 | 98.96 | 4153 | Alpine meadow | −5.1 | 0.36 | −1.62 | 2.5 | 2014–2021 | |
| PT9 | 38.63 | 98.95 | 3970 | Alpine swamp meadow | −4.2 | 0.50 | −1.38 | 1.9 | 2014–2021 | |

## 4.2 Snow depth correction

The snow depth from ERA5-Land was reported to be over biased due to the significant drawback of snow densification and precipitation represented in models (Cao et al., 2020, 2022; Orsolini et al., 2019). To reduce the uncertainty and to algin with FPM, the snow depth was derived from ERA5-Land snow water equivalent by dividing the static snow density of $250\,\mathrm{kg\,m^{-3}}$. The corrected ERA5-Land snow depth was reduced by about $2\,\mathrm{cm}$ (or 18%) by mean (Fig. D2).

## 4.3 Surface cover

FPM considers the influences of vegetation on permafrost via the processes of latent heat and soil moisture etc. (Appendix A). In FPM, static vegetation is assumed and the vegetation optical depth (VOD), leaf area index (LAI), and vegetation type are required (Table 1). For snow-free periods, the ground albedo is from Jia et al. (2022).

The remote-sensing datasets are different in temporal coverage, so we use the climatology to represent the long-term conditions. The median of the daily (VOD, albedo, and $\theta_{\mathrm{R}}$) or monthly (LAI) mean during the coverage period was used. All the input surface products are bi-linearly interpolated to the simulation grids.

## 4.4 Datasets used for model evaluation

### 4.4.1 Synthesis datasets

In this study, 10 synthesis sites with both meteorological and soil temperature measurements were used to conduct the detailed evaluations (Table 3, Fig. 1). The soil information, i.e., soil texture and moisture are also available at the sites. Note that







**Figure 2.** Comparison of simulated active layer soil temperature with time series at the synthesis sites. The daily soil temperature present are averaged based on all available sites and years for each vegetation type and different soil depths, and the numbers of sites are given in parentheses. Observations are in black, red lines show the simulation forced by reanalyses, and the blue lines represent that forced by observed atmospheric forcing and *in situ* soil information (if available). The shaded areas depict the ensemble range from the 25th to 75th. The ensemble of observation forced simulation are produced using results from different sites and additional ranges of soil moisture (see Table 2).




the missing atmospheric observations are filled with downscaled reanalyses following the algorithm presented by Fiddes and Gruber (2014); Cao et al. (2017a).

### 4.4.2 Active layer thickness and borehole temperature

Active layer thickness and borehole temperature measurements from the Global Terrestrial Network for Permafrost (GTN-P, Biskaborn et al., 2019) and literature (see Supplement for details) are used here to evaluate model performance. This yielded 247 ALT measurements from 128 sites (Fig. 1). The measured mean ALT was about $2.3 \pm 0.8$ m with a range of 0.7–4.9 m. ALTs are derived from different land covers, including Alpine desert, Alpine steppe, Alpine meadow, and Alpine swamp meadow, indicating that the evaluation presented here is representative.

The MAGT measurements between the depths of 3 and 40 m are treated as the thermal state of permafrost, and this leads to 70 boreholes with 241 MAGT measurements that were used for model evaluation. The measured mean MAGT was about $-0.91 \pm 0.8$ °C with a range from $-3.3$ to 1.9 °C. Among the measurements, 18 borehole sites with a long times-series ($\geq 1$ decade) were further used to evaluate the modeled TSP changes.

### 4.4.3 Referenced permafrost distribution

We use five permafrost maps as reference datasets to evaluate the modeled permafrost area. These are (1) the new map of permafrost distribution on the TP via the semi-physical 'temperature at the top of permafrost' (TTOP) model (Zou et al., 2017); (2) the permafrost zonation index map compiled based on the statistical relationship between topoclimatic predictors (e.g., air temperature, snow, and vegetation) and permafrost zonation (Cao et al., 2019b); (3) the global permafrost zonation index with the normal and cold variant (Gruber, 2012); (4) the Northern Hemisphere permafrost map derived via the TTOP model (Obu et al., 2019), and (5) the outputs from LSM of Noah-MP (Wu et al., 2018). The permafrost extent from these maps is estimated for different time periods using different modeling paradigms and are not perfect or a 'source of truth'. We treat them as an ensemble and use their range and mean as the 'best available' reference.

## 5 Results

### 5.1 Model evaluations

Our evaluation results showed the overall RMSE of daily soil temperature in the active layer was 2.1 °C with a BIAS of 0.2 °C (Figure 2). FPM showed relatively worse performance in areas with alpine swamp meadow (RMSE = 3.6 °C), with warm bias in summer and cold bias in winter. This is attributed the poorly prescribed soil information, i.e., peat layer and soil moisture. To demonstrate the hypothesis, we conducted the additional simulations using observed atmospheric forcing and soil profile from borehole measurements. The simulated soil temperature was significantly improved by 2.1 °C, indicating FPM could be improved with more reliable climate forcing and soil profile (Fig. 2).



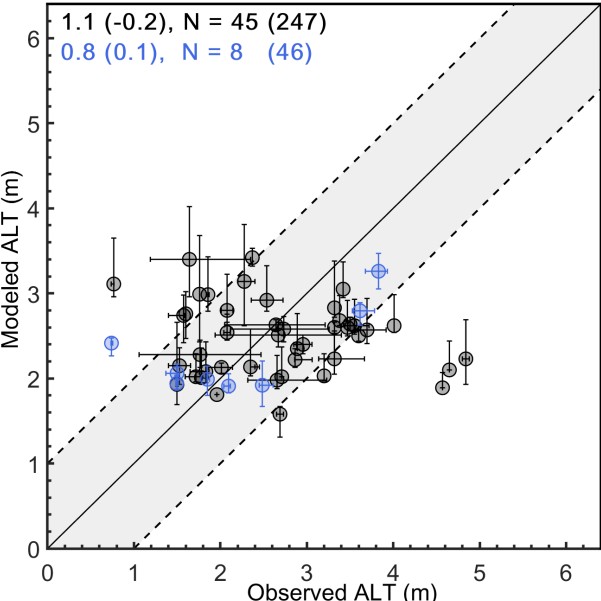

**Figure 3.** Evaluation of modeled active layer thickness (ALT). The values are root-mean-square-error and bias (in parentheses). N represents the number of grids used for evaluation after aggregating sites in the same grid, and the number of observations is given in parentheses. The ensemble mean from FPM simulations are given in black dot and the range between the 25[th] and 75[th] percentiles are given in whiskers. The additional simulation driven by observed meteorological forcing are given in blue. Dashed lines indicate $\pm$ 1 m.

FPM generally has good agreement with observed ALTs with the overall RMSE of 1.1 m, and slightly underestimated ALT (BIAS = $-0.2$ m) due to the cold-biased summer air temperature (Fig. 3 and D1). Following the relatively worse soil temperature, ALT was over-biased in areas with alpine swamp meadow. The additional simulation with observed forcing and

soil information, again, showed much more promising results (RMSE = 0.8 m). The ALT bias was within $\pm$ 1 m at most (62 %) evaluated cells (Fig. 3).

FPM is found to underestimate the MAGT with an overall mean BIAS of $-1.1$ °C, which is aligned with the cold-biased air temperature (Fig. D1a and d). The overall MAGT RMSE was 1.5 °C (Fig. 4), and about 48 (81)% sites have a bias within $\pm$ 1 (2) °C. Although the MAGT change trend is well addressed by FPM with an RMSE of 0.21 °C dec$^{-1}$, it is found even

greater than the observed mean MAGT trend of $0.12 \pm 0.09$ °C dec$^{-1}$ (Fig. 5). This indicates that the simulated MAGT trend may not be reliable. In fact, the permafrost warming at the measured sites was relatively gentle (with a range from $-0.07$ to 0.3 °C dec$^{-1}$) compared to that in high latitudes (Hock et al., 2019). This is because the permafrost temperature over the TP is very "warm", and the heat from atmosphere was consumed by phase change rather than temperature increase (also see Fig. 7). The significant latent heat introduce additional challenge for reproducing MAGT trend.

Excluding glaciers and lakes, the estimated current (2010–2023) permafrost area was about $1.15 \pm 0.02 \times 10^6$ km$^2$ based on the ensemble simulations from FPM. This is found reasonable compared to the referenced ensemble mean of $1.15 \pm 0.12$





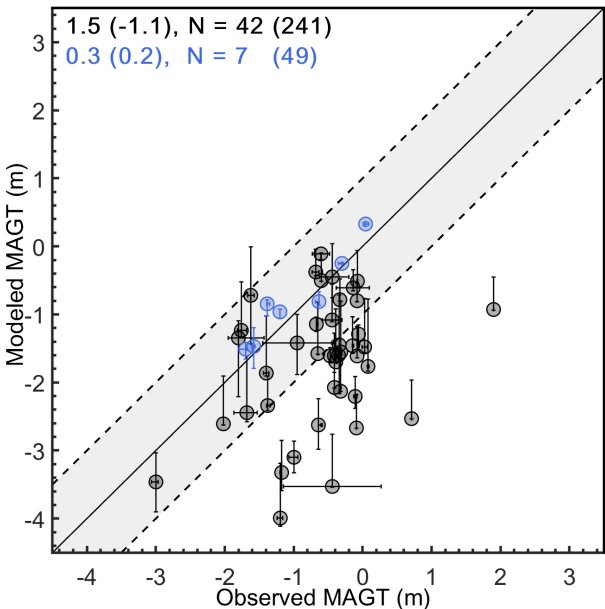

**Figure 4.** Same as Figure 3, but for the mean annual ground temperature (MAGT). Dashed lines indicate $\pm$ 1 °C.

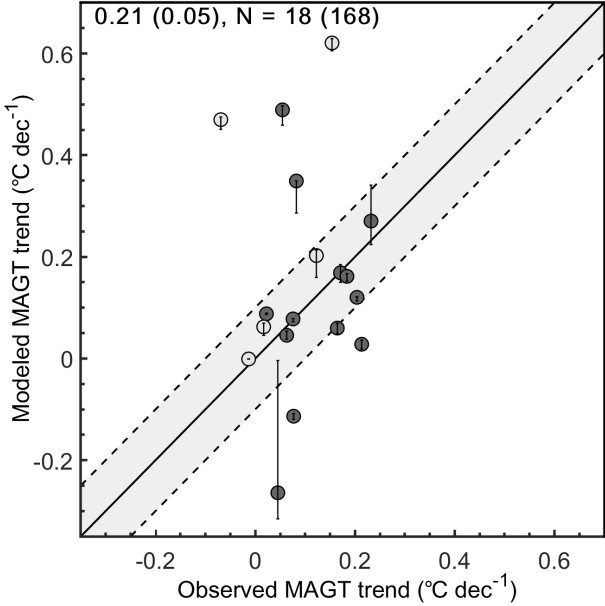

**Figure 5.** Same as Figure 3, but for the mean annual ground temperature (MAGT) changes. Only the sites with long-term observations ($\geq$ 1 decade) are used here. The filled dots are sites with observed significant trends. Dashed lines indicate $\pm$ 0.1 °C dec$^{-1}$.



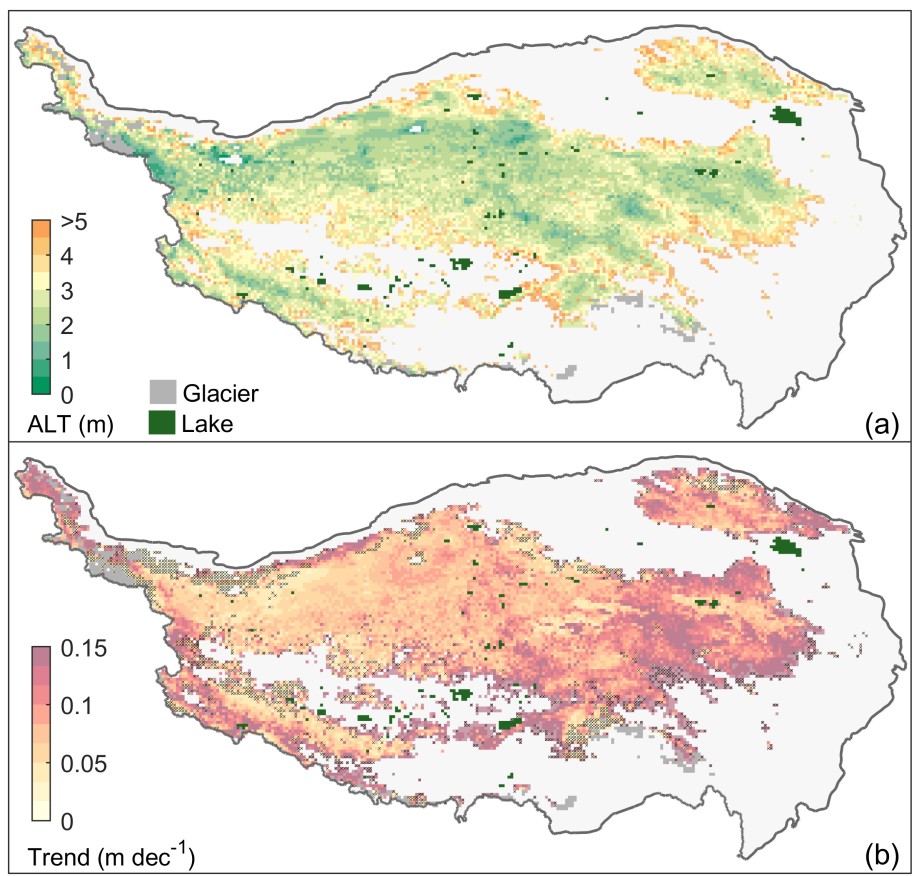

**Figure 6.** (a) The modeled current (2010–2023) mean active layer thickness (ALT), and (b) its change rate from 1980 to 2023. The areas without significant trends are marked by ×. Glaciers and lakes are from ERA5-Land.

$\times\,10^6\,\mathrm{km}^2$ (Fig. 8). In addition, our model performance and simulated ALT, MAGT, and permafrost extent are comparable to the stand-alone model of CryoGridLite (Chen et al., 2025).

**5.2 Changes in active layer thickness**

The ensemble simulations from FPM showed the current (2010–2023) overall mean ALT was about $2.88 \pm 0.95$ m over the TP. The results are found highly align with the geothermal model, i.e., 2.1–2.4 m, from Qin et al. (2017). Our results indicated that about 34.1 % of permafrost regions have an ALT greater than 3 m, highlighting that the widely used land surface models and reanalyses with shallow soil column may not be sufficient for permafrost studies over the TP. The long-term simulation showed that ALT had a inter-annual trends with a decreasing trend between 1950 and 1980 ($-0.06\,\mathrm{m\,dec^{-1}}$), followed by

a dramatically continuous increasing trend ($0.07\,\mathrm{m\,dec^{-1}}$). Consequently, the ALT over TP increased by 0.30 m since 1980 (Fig. 6b).



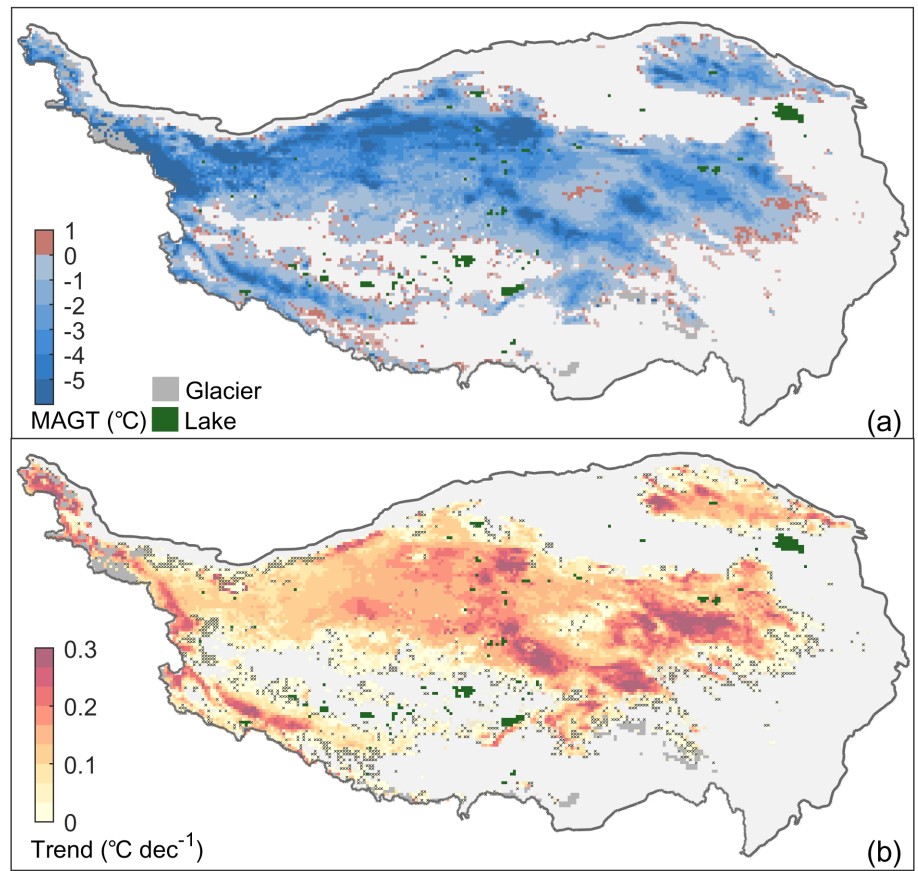

**Figure 7.** (a) The modeled current (2010–2023) mean annual ground temperatures at the depth of 15 m, and (b) its change rate from 1980 to 2023. The areas without significant trends are marked by ×.

## 5.3 Changes in permafrost temperature

The mean annual ground temperature at the depth of 15 m (MAGT$_{15}$) is used to represent the thermal state of permafrost. The modeled overall mean MAGT$_{15}$ (2010–2023) was about $-2.0 \pm 2.1$ °C for the permafrost regions over the TP and shows a wide range from $-19.1$ to $2.7$ °C (Fig. 7a). Our simulations revealed that the overall MAGT$_{15}$ increased by approximately 0.17 °C since 1950, with a more dramatic warming of 0.26 °C (or 0.06 °C dec$^{-1}$) since 1980. Similar to the ALT, MAGT shows a clear cooling trend ($-0.03$ °C dec$^{-1}$) between 1950 and 1980 corresponding to the changes in near surface air temperature.

## 5.4 Changes in permafrost extent

The permafrost area decreased by about $6.7 \times 10^4$ km$^2$ dec$^{-1}$ during 1950–2023, but increased by 5.2 % from 1950 to 1980 following the cooling of near-surface air temperature (Fig. 9b). Permafrost area decreased at a rate of $3.0 \times 10^4$ km$^2$ dec$^{-1}$, or a total area of 10.5 %, since 1980. Our results showed that the model with shallow soil column would significantly underestimate



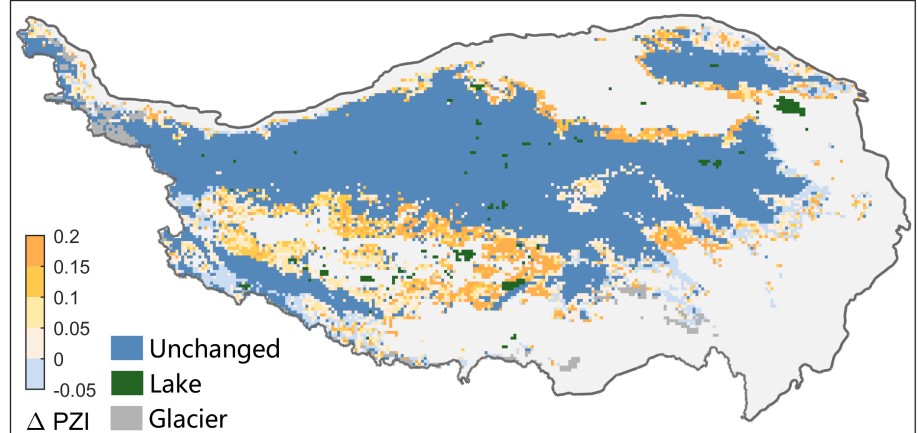

**Figure 8.** Changes of permafrost extent between 1991–2020 and 1951–1980. The permafrost zonation index (PZI) are 45-member ensemble probability of permafrost where permafrost is defined by the daily temperature at 15 m depth.

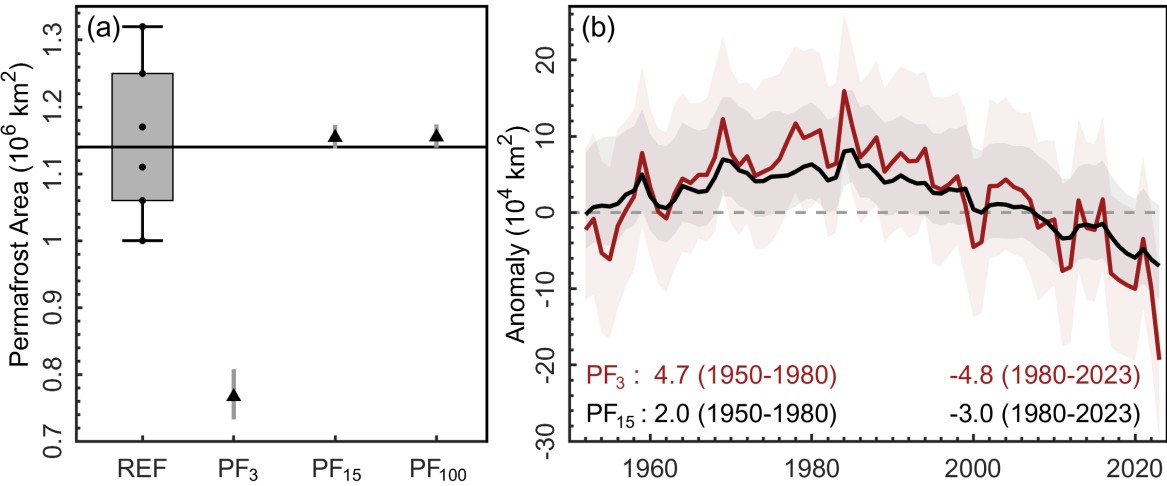

**Figure 9.** (a) Comparison of estimated current (2010–2023) permafrost area from FPM simulations with referenced estimations, and (b) anomaly of permafrost area since 1950. The subscript means the depth above which permafrost is diagnosed. The permafrost area trend ($10^4 \, \mathrm{km^2 \, dec^{-1}}$) is estimated for the periods before and after 1980, separately.

permafrost area but overestimated permafrost degradation. Take the top 3 m as an example, which has been widely used in the land surface model. The estimated near-surface (top 3 m) permafrost area ($7.67 \times 10^4 \, \mathrm{km^2}$) was about 33.4 % smaller compared to the ground "truth", or 33.6 % smaller than the simulations with sufficient soil column (e.g., 100 m, Fig. 9a). 295 In addition, the permafrost degradation was overestimated by about 60 % ($3.0 \times 10^4 \, \mathrm{km^2 \, dec^{-1}}$ *vs*. $4.8 \times 10^4 \, \mathrm{km^2 \, dec^{-1}}$) with shallow soil column (Fig. 9). This highlights that the current land surface models with shallow soil column can lead to significant uncertainties in permafrost simulations.





## 6 Discussions

### 6.1 Uncertainties from climate forcing and input datasets

We used the ERA5-Land reanalyses as model forcing. However, the ERA5-Land is found cold biased in near-surface air temperature (Appendix D, Fig. D1), leading to underestimated ALT as well as MAGT (Fig. 6 and 7). In fact, permafrost simulations are hampered by reduced reanalyses quality in cold regions primarily due to inherent challenges in representing nonlinear processes involving ice, or its phase change near 0 °C (Cao and Gruber, 2025). The poorly described soil column, especially the soil organic matter, put additional uncertainty for permafrost simulations.

### 6.2 Model limitations

In this study, we introduce and demonstrated the suitability of FPM for large-scale permafrost simulation. For the initial version of FPM, there are four main limitations. First, the snow scheme is simplistic, although significant influences of snow cover on soil thermal regime have been well documented (Zhang, 2005). FPM currently does not consider the snow mass balance, and, therefore, additionally requires snow depth as input. The static snow density was used to represent the overall conditions
during the snow-covered period. We clarify that the parameter used essentially overestimated the density for fresh snow but underestimated the old snow. Further, the snow cover fraction used here was developed for high latitudes and does not consider the influences of complex terrain, which may redistribute the snow cover and lead to strong heterogeneity for the soil thermal regime. While the simple snow scheme seems adequate for the Tibetan Plateau with little snow (Fig. D2), considerable efforts are required to improve snow processes if FPM would applied to snow-prevalent areas, i.e., the high latitudes.

Second, FPM does not consider the water balance, and the static surface conditions, including vegetation and albedo (varies with season but remains unchanged among years), were assumed via using the remote-sensing-based climatology. This means the influences of surface and sub-surface changes are not accounted for and limited the model ability in predicting long-term permafrost changes.

Third, the fine-scale influences (i.e., debris and peat layer) are either not or simply represented here due to the simulation
scale (∼10 km). This may overestimate permafrost degradation, especially for the areas near the permafrost lower limit, where the relict permafrost was found (Fig. 8, Cao et al., 2021).

Fourth, FPM does not consider the thermokarst processes or the so-called "abrupt thaw" raised by excess ice loss. The thermokarst was thought as local-scale tipping element that would remarkably accelerate permafrost degradation (Devoie et al., 2019).

### 6.3 Comparison with other permafrost models

Our results are found comparable to previous simulations derived from the geothermal numerical models (e.g., Qin et al., 2017; Chen et al., 2025) as well as land surface models (e.g., Guo and Wang, 2013; Wu et al., 2018; Zheng et al., 2020). FPM, as the stand-alone permafrost model, benefits from the consideration of land-atmosphere processes (i.e., the surface energy





balance and vegetation effects) typically not included in geothermal numerical models, while maintaining the deep soil column
and suitable numerical solver for soil phase change. In addition, the land-surface-scheme designed structure and streamlined
processes (i.e., its efficient simulation of latent heat) make it suitable for large-scale ensemble simulations. Different from the
other land surface scheme models with rich processes beyonds permafrost, such as SUTRA (McKenzie et al., 2007), Advanced
Terrestrial Simulator (Jan et al., 2020), and CryoGrid 3 (Westermann et al., 2023), which are applied to fine scales, the initial
FPM aim to provide a flexible yet simple platform for large-scale permafrost simulation studies.

### 6.4  Future developments

Previous studies reported that the permafrost processes of the Earth system model in CMIP6 is limited improved compared to
the previous generation of CMIP5 (e.g., Burke et al., 2020), and current Earth system models generally have weak ability in
representing permafrost (Schädel et al., 2024). This highlights the urgent need to develop the stand-alone permafrost model.
Since the required inputs are derived from global datasets, this opens the opportunity for global permafrost simulation with the
FPM platform. The incorporation of a state-of-the-art snow scheme – particularly critical for high-latitude permafrost processes
– will further enhance this capability. We hope with improved permafrost ground ice maps and a rigorously validated solution
(addressing both physical processes and numerical solver aspects), we can implement excess ice loss processes within FPM to
represent permafrost "abrupt thaw".

### 7  Conclusions

In this study, we introduce a new land surface scheme specifically designed for permafrost applications, the Flexible Permafrost
Model (FPM). This model serves as a flexible platform to explore novel parameterizations for a variety of permafrost processes.
To demonstrate the utility of FPM for supporting permafrost studies, we apply the model to permafrost studies over the Tibetan
Plateau. Our simulation results are compared to *in situ* observations and published permafrost extent. We summarize the main
contributions and insights of this work as the following:

1. FPM shows suitability in reproducing permafrost characteristics, such as active layer thickness, and the thermal state.
With more reliable inputs, especially soil profile, FPM-based simulations can be further improved;

2. Simulations indicated that the current (2010–2023) mean active later thickness was about $2.88 \pm 0.95$ m, permafrost
temperature at a depth of 15 m was about $-2.0 \pm 2.1\,^{\circ}\text{C}$, and permafrost extent was about $1.15 \pm 0.02 \times 10^6\,\text{km}^2\,\text{dec}^{-1}$
over the Tibetan Plateau;

3. The process-based historical simulation revealed steady permafrost degradation over the Tibetan Plateau since 1980.
The active layer thickness increased by 0.30 m, permafrost temperature at 15 m depth increased at a rate of $0.06 \pm 0.01$
$^{\circ}\text{C}\,\text{dec}^{-1}$, and permafrost extent degraded by about 10.5 %;

4. Our simulations indicate that current land surface models employing shallow soil columns are inadequate for permafrost
research on the Tibetan Plateau, since they have generally underestimated permafrost extent while overestimating degra-



dation rates. Such inadequacy may also pose challenges in other regions characterized by deep active layers (i.e., $> 3$ m);

5. This study highlights the ongoing efforts in stand-alone process-based permafrost model development. We hope in the future, with more available stand-alone land-scheme-designed permafrost models, the permafrost community will provide a simulation benchmark for the Earth system model developments as well as the climate change assessment at a

global scale.



## Appendix A: Latent heat from Priestley-Taylor method

The latent heat flux ($Q_e$) is treated differently depending on the snow cover, and the FMP uses the Priestley-Taylor method by generally following Martens et al. (2017). If snow is absent, $Q_e$ is the sum of latent heat for bare soil $Q_e^s$ ($\mathrm{W\,m^{-2}}$) and covered vegetation $Q_e^v$ ($\mathrm{W\,m^{-2}}$).

$$Q_e = Q_e^s + Q_e^v \tag{A1}$$

where $Q_e^s$ and $Q_e^v$ are given as

$$Q_e^s = S_s \alpha_{pt} \frac{\Delta(Q_n^s - Q_c)}{\Delta + \gamma} \tag{A2}$$

$$Q_e^v = S_v \alpha_{pt} \frac{\Delta Q_n^v}{\Delta + \gamma} \tag{A3}$$

where $Q_n^s$ and $Q_n^v$ are the net radiation partitioned for bare soil and vegetation, and derived following Fisher et al. (2008).

$$Q_n^s = f_s Q_n \tag{A4}$$

$$Q_n^v = (1 - f_s) Q_n \tag{A5}$$

where $Q_n$ ($\mathrm{W\,m^{-2}}$) is the total net radiation, $f_s$ ($-$) is fractional net radiation reached to the bare soil, and is treated as

$$f_s = \exp(-k_{Rn} LAI) \tag{A6}$$

where $k_{Rn} = 0.6$ is extinction coefficient from Impens and Lemeur (1969), and LAI ($-$) is Leaf Area Index from MODIS (Table 1).

While $\alpha_{pt}$ is normally set to 1.26 for snow-free areas following the original study (Priestley and Taylor, 1972), many studies reported that it is site-specific, depending on the surface conditions (e.g., Constantin et al., 2015; Martens et al., 2017). In FPM, the parameterization scheme that solves the influences of vegetation and soil moisture on latent heat was used. This is achieved via introducing the evaporation stress factor S ($-$). The S for bare soil ($S_s$) and vegetation ($S_v$) are defined as:

$$S_s = 1 - \frac{\theta_c - \theta_R}{\theta_c - \theta_r} \tag{A7}$$

$$S_v = \sqrt{\frac{VOD}{VOD_{max}}} \left(1 - \left(\frac{\theta_c - \theta_R}{\theta_c - \theta_{wp}}\right)^2\right) \tag{A8}$$

where $\theta_c$ ($\mathrm{m^3\,m^{-3}}$) is the critical soil moisture, $\theta_R$ ($\mathrm{m^3\,m^{-3}}$) is the soil moisture in the root zone layer, $\theta_r$ ($\mathrm{m^3\,m^{-3}}$) is the residual soil moisture, $\theta_{wp}$ ($\mathrm{m^3\,m^{-3}}$) is soil moisture content of wilting point, VOD ($-$) is vegetation optical depth, and $VOD_{max}$ is the maximum VOD for a given simulation cell. The $\theta_c$ is variable depending on the soil texture, and we used the parameters from Zhu et al. (2019). The $\theta_R$ ($\mathrm{m^3\,m^{-3}}$) is estimated as the ensemble mean of five remote-sensing-based soil moisture (Table 1), $\theta_{wp}$ is given in Appendix B, and $\theta_r$ is constant of 5 $\mathrm{m^3\,m^{-3}}$ following Fisher et al. (2008). The VOD is from Moesinger et al. (2020).



If snow is present, the $Q_e$ for a given cell is derived as the weighted mean of the snow and soil cover fractions (SCF), assuming no latent heat via vegetation:

$$Q_e = SCF \cdot Q_e^{sn} + (1 - SCF) \cdot Q_e^{s} \tag{A9}$$

where $Q_e^{sn}$ is the latent heat from snow-covered area, and is given as

$$Q_e^{sn} = S_{sn} \alpha_{pt} \frac{\Delta (Q_n^{sn} - Q_c)}{\Delta + \gamma} \tag{A10}$$

where $S_{sn}$ and $\alpha_{pt}$ are set to 1 and 0.95, respectively.

## Appendix B: Soil organic matter properties

FPM considers the impacts of organic matter on soil hydraulic properties following CoLM (Dai et al., 2003) and SURFEX (Masson et al., 2013):

$$\tau = f_o \tau^o + f_m \tau^m + f_g \tau^g \tag{B1}$$

$$f_o + f_m + f_g = 1 \tag{B2}$$

where $f_o$ ($m^3\,m^{-3}$), $f_m$ ($m^3\,m^{-3}$), and $f_g$ ($m^3\,m^{-3}$) are the fractions of organic matter, mineral and gravel, respectively. The $\tau$ refers to the soil hydraulic properties, including: slope of the retention curve b (dimensionless), soil matric potential $\psi$ (dimensionless), soil porosity $\phi$ ($m^3\,m^{-3}$), field capacity $\theta_{fc}$ ($m^3\,m^{-3}$), and wilting point $\theta_{wp}$ ($m^3\,m^{-3}$). For gravel, $\phi$ is set as $0.05\,m^3\,m^{-3}$, and others are set to 0, assuming the soil hydraulic properties are not directly affected by gravel.

The soil organic matter properties $\tau^o$ were implicitly accounted for in the FPM:

$$b^o = min\left(2.7 + 9.3 \times \frac{Z_i}{Z_{sap}}, \, 12.0\right) \tag{B3}$$

$$\psi_{sat}^o = max\left(10.3 - 0.2 \times \frac{Z_i}{Z_{sap}}, \, 10.1\right) \tag{B4}$$

$$\phi^o = max\left(0.93 - 0.1 \times \frac{Z_i}{Z_{sap}}, \, 0.83\right) \tag{B5}$$

$$\theta_{fc}^o = min\left(0.37 + 0.35 \times \frac{Z_i}{Z_{sap}}, \, 0.72\right) \tag{B6}$$

$$\theta_{wp}^o = min\left(0.07 + 0.15 \times \frac{Z_i}{Z_{sap}}, \, 0.22\right) \tag{B7}$$

where $Z_i$ (m) is the soil depth for soil grid of i, and $Z_{sap} = 0.5$ m is the depth that organic matter takes on the characteristics of sapric peat.



The mineral soil hydraulic properties were approximated as:

$$b^m = 2.91 + 15.9 f_{clay} \tag{B8}$$

$$\psi_{sat}^m = -10 \times 10^{(1.88 - 1.31 f_{sand})} \tag{B9}$$

$$\phi^m = 0.489 - 0.126 f_{sand} \tag{B10}$$

$$\theta_{fc}^m = 0.45 \left( f_{clay} \right)^{0.3496} \tag{B11}$$

$$\theta_{wp}^m = 0.37 \sqrt{f_{clay}} \tag{B12}$$

where $f_{clay}$ ($\mathrm{m^3\,m^{-3}}$) and $f_{sand}$ ($\mathrm{m^3\,m^{-3}}$) are the volumetric content of clay and sand to the total mineral soil, respectively.

## Appendix C: Evaluation metrics

The evaluation metrics of bias (BIAS) and root-mean-square-error (RMSE) were used here to evaluate model performance.

$$BIAS = \frac{1}{N} \sum_{i=1}^{N} (MOD - OBS), \tag{C1}$$

$$RMSE = \sqrt{\frac{1}{N} \sum_{i=1}^{N} (MOD - OBS)^2}, \tag{C2}$$

where $OBS$ and $MOD$ represent the variables of interest (i.e., mean annual ground temperature and active layer thickness) from *in situ* observations and the simulations.

## Appendix D: Evaluation of model forcing

The climate forcing was evaluated against *in situ* observations from the synthesis sites (Table 3). Our evaluation results indicate that there is generally a cold bias of the near-surface air temperature ($-1.9\ ^\circ\mathrm{C}$) for the ERA5-Land, and the RMSE was about $3.5\ ^\circ\mathrm{C}$ (Fig. D1). While ERA5-Land slightly overestimated the incoming short-wave radiation (bias = $10.3\ \mathrm{W\,m^{-2}}$), the incoming long-wave radiation was underestimated by about $-24.1\ \mathrm{W\,m^{-2}}$. Overall, the evaluation results indicate that the meteorological variables in the reanalysis dataset are generally consistent with the observations and can be used as suitable forcing/inputs for the numerical simulation.

## Appendix E: Nomenclature

In this study, Q ($\mathrm{W\,m^{-2}}$) refers to surface energy balance terms, subscripts identify the term (si: incoming shortwave radiation; li: incoming longwave radiation; le: emitted longwave radiation; h: turbulent exchange of sensible heat; e: turbulent exchange of latent heat; c: energy transport due to conduction; n: net radiation; and m: the energy flux available for melt). We use k ($\mathrm{W\,m^{-1}\,^\circ C^{-1}}$), CV ($\mathrm{J\,m^{-3}\,^\circ C^{-1}}$), and $\rho$ ($\mathrm{kg\,m^{-3}}$) to represent thermal conductivity, volumetric heat capacity, and density,




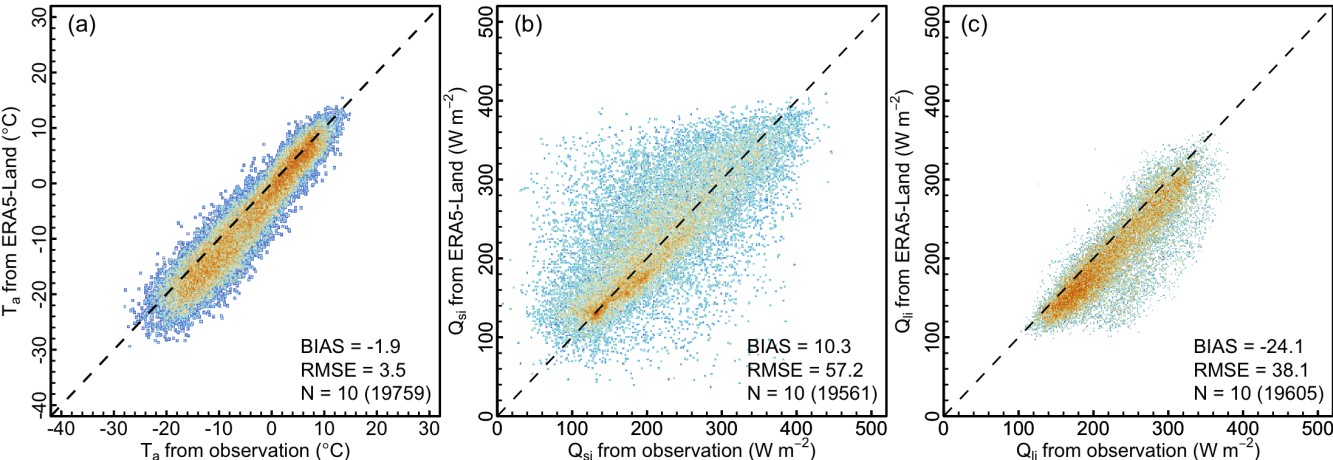

**Figure D1.** Evaluation of the daily (a) near-surface air temperature ($T_a$), (b) incoming short-wave radiation ($Q_{si}$), and (c) incoming long-wave radiation ($Q_{li}$) from ERA5-Land. The number of meteorological stations (measurements) used for the evaluation are given as N.

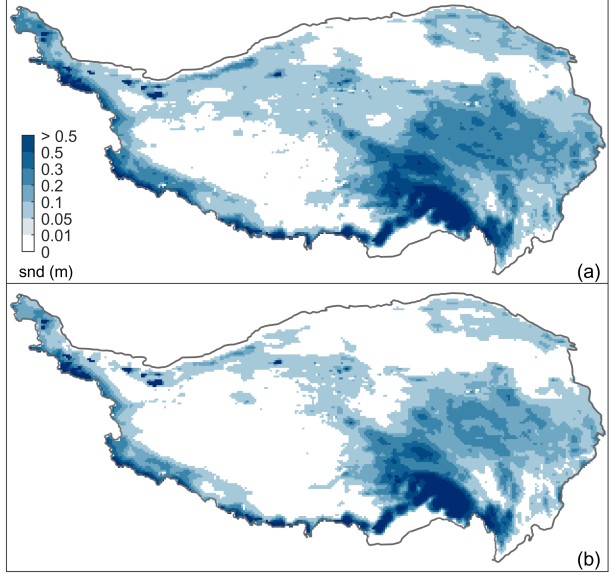

**Figure D2.** The mean winter (DJF) snow depth (1991–2020) from ERA5-Land (a) and its correction with a static snow density of 250 $\mathrm{kg\,m^{-3}}$ (b).

respectively. The subscript identifies each source (s: soil; sn: snow; m: mineral; o: organic; w: water; i: ice; a: air; and g: gravel). The volumetric contents for each soil component is represented by f ($\mathrm{m^3\,m^{-3}}$).



**Table D1.** Ground layer depths (m) and thicknesses (m) for the FPM ground layers configuration.

| Depth | Thickness | Layer |
|-------|-----------|-------|
| 0–0.2 | 0.01 | 1–20 |
| 0.2–1 | 0.05 | 21–36 |
| 1–5 | 0.10 | 37–76 |
| 5–10 | 0.20 | 77–101 |
| 10–20 | 0.50 | 102–121 |
| 20–50 | 1.00 | 122–151 |
| 50–150 | 5.00 | 151–160 |

*Code availability.* The Flexible Permafrost Model (FPM) source code is available on request from Bin Cao (bin.cao@itpcas.ac.cn).

*Data availability.* The active layer thickness, mean annual ground temperature at 15 m depth, and the permafrost extent–as the 45-member
ensemble mean–are publicly available via Zenodo (https://10.5281/zenodo.15229474; Sun and Cao, 2025).

*Author contributions.* WS carried out this study by developing model, analyzing data, performing the simulations, and writing the paper and
was responsible for the compilation and quality control of the observations. BC conceived and guided the project, proposed the initial idea,
designed model structure as well as parameterizations, developed and tested the model, and contributed to organizing as well as writing the
paper.

*Competing interests.* The authors declare that they have no conflict of interest.

*Acknowledgements.* The authors thank Kun Zhang for his helpful suggestions in model development and Shengdi Wang for developing and
testing the early version of model. ERA5-Land reanalysis data and the ESA CCI LC map are provided by the ECMWF.

This study was supported by the National Natural Science Foundation of China (grant no. 42422608), the Youth Innovation Promotion As-
sociation Chinese Academy of Sciences (grant no. 2023075) to B. Cao. W. Sun was supported by the China Postdoctoral Science Foundation
(grant no. 2023M733604).



**Table E1.** Nomenclature and input parameters for Flexible Permafrost Model (FPM).

| Symbol | Parameter | Value or range | Unit |
|---|---|---|---|
| $C$ | apparent heat capacity | | $\mathrm{J\,m^{-3}\,K^{-1}}$ |
| $L$ | volumetric latent heat of fusion for ice | | $\mathrm{J\,m^{-3}}$ |
| $\theta_u$ | volume contents of unfrozen water | | $\mathrm{m^3\,m^{-3}}$ |
| $\theta_i$ | volume contents of ice | | $\mathrm{m^3\,m^{-3}}$ |
| $\theta_a$ | volume contents of air | | $\mathrm{m^3\,m^{-3}}$ |
| $\theta_R$ | soil moisture in root zone | | $\mathrm{m^3\,m^{-3}}$ |
| $\theta_{wp}$ | soil moisture content of wilting point | | $\mathrm{m^3\,m^{-3}}$ |
| $\theta_v$ | soil moisture in vadose zone | | $\mathrm{m^3\,m^{-3}}$ |
| $\theta_{sat}$ | saturated soil moisture | | $\mathrm{m^3\,m^{-3}}$ |
| $\theta_r$ | residual soil moisture | | $\mathrm{m^3\,m^{-3}}$ |
| $\theta_{fc}$ | soil field capacity | | $\mathrm{m^3\,m^{-3}}$ |
| $\phi$ | soil porosity | | $\mathrm{m^3\,m^{-3}}$ |
| $\alpha$ | surface albedo | | Dimensionless |
| $\alpha_g$ | snow-free surface albedo | | Dimensionless |
| $\alpha_{sn}$ | snow albedo | 0.50–0.85 | Dimensionless |
| $\alpha_{sn}^{max}$ | maximum snow albedo | 0.85 | Dimensionless |
| $\alpha_{sn}^{min}$ | minimum snow albedo | 0.50 | Dimensionless |
| $T_a$ | near-surface air temperature | | K |
| $T$ | ground or/and snow temperature | | K |
| $T_{s0}$ | ground or snow surface temperature | | K |
| $Z$ | total depth of the analysis domain | | m |
| $D_h$ | exchange coefficients for heat | | Dimensionless |
| $S$ | evaporation stress factor | | Dimensionless |
| $\alpha_{pt}$ | Priestly-Taylor coefficient | | Dimensionless |
| $\Delta$ | slope of the saturation vapor pressure temperature curve | | $\mathrm{Pa\,K^{-1}}$ |
| $\gamma$ | psychrometric constant | | $\mathrm{Pa\,K^{-1}}$ |
| $e_a$ | atmospheric vapor pressure | | Pa |
| $e_s$ | snow or soil surface vapor pressure | | Pa |
| $\epsilon_s$ | surface emissivity | | Dimensionless |
| $P_a$ | atmospheric pressure | | Pa |
| $u_z$ | wind speed | | $\mathrm{m\,s^{-1}}$ |
| $z_0$ | roughness length | | m |
| $\sigma$ | Stefan-Boltzmann constant | $5.67 \times 10^{-8}$ | $\mathrm{W\,m^{-2}K^{-4}}$ |
| $\kappa$ | Von Karman's constant | 0.4 | Dimensionless |
| $L_f$ | mass specific latent heat of water | $0.334 \times 10^6$ | $\mathrm{J\,kg^{-1}}$ |
| $L_v$ | latent heat of vaporization | $2.471 \times 10^6$ | $\mathrm{J\,kg^{-1}}$ |
| $c_p$ | specific heat of air | 1004.0 | $\mathrm{J\,K^{-1}\,kg^{-1}}$ |
| $\rho_{sn}$ | density of the snow | 250 | $\mathrm{kg\,m^{-3}}$ |



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
