# Peer review of "Ensemble numerical simulation of permafrost over the Tibetan Plateau from Flexible Permafrost Model: 1950–2023"

_EGUsphere, 2025_

## Referee Comment (RC1)

Permafrost underlies roughly 15% of the Northern Hemisphere's exposed land. Its thaw is reshaping hydrology and ecosystems and undermining the stability of infrastructure. Understanding the trajectory of permafrost dynamics under continued warming is therefore essential. At regional-to-hemispheric scales, numerical models are essential for reconstructing past states, attributing observed trends, and projecting future permafrost dynamics. Here, Sun and Cao introduce the Flexible Permafrost Model (FPM), a standalone land-surface scheme configured in one-dimensional heat conduction, and apply it to a long (1950–2023) ensemble simulation over the Tibetan Plateau (TP). The experiments are forced by ERA5-Land reanalysis data and use a deep soil column (150 m) to reconstruct the permafrost thermal regime. A 45-member ensemble to represent the broad uncertainty of hydrological parameters, and yields spatially consistent estimates of active-layer thickness (ALT), mean annual ground temperature (MAGT), and permafrost areas with observed and previous studies. Evaluation against site observations shows skill of the correct order of magnitude, and the experiments clarify how shallow-column diagnostics can bias permafrost area and trend estimates relative to deep-column simulations.

Overall, this work should be considered by The Cryosphere, provided that the authors address the comments below and supply the requested clarifications.

Comments:

1. The author stated that the geothermal numerical model lacks the link with the atmosphere, and the land surface models are not good at representing the permafrost processes. FPM coupled the advantages of these two models to deal with the land-atmosphere interactions and extend the soil column more deeply. Please articulate the specific advantages of FPM relative to existing models: e.g., demonstrably higher accuracy, computational efficiency, or novel parameterizations that capture landscape dynamics.

2. Lines 5–6: The author states, "The FPM accounts for both vertical and lateral heat flow ...". Yet the present application appears strictly 1-D. Please make this distinction explicit in the Abstract/Introduction/Methods to avoid implying that lateral heat-flux parameterizations are active in this study, or provide details if they are.

3. Section 3.3: I was wondering do the ensemble parameters come from both Table 1 and Table 2 (Lines 177–178), or only from the hydrological parameters in Table 2? In addition, please justify the choice of 45 members per grid cell.

4. Line 190: Could you clarify the spin-up convergence criterion? For example, which variables were evaluated, and what thresholds or tolerances were applied to judge convergence? I also suggest considering a dynamic spin-up for regional runs, which may be more efficient than a fixed 1000-year spin-up per grid cell.

5. Figure 2: First, please add a legend. It is hard for me to recognize the meaning of the different lines. Second, reanalysis forced simulations exhibit a larger seasonal amplitude (colder winters, warmer summers) than observations; however, it seems cannot be explained by the cold bias of reanalysis forcing. Third, are simulation–observation comparisons shown for the same calendar year for all subfigures? If so, which year? Fourth, could the author explain the meaning of the red and blue numbers? I assume they report RMSE and BIAS for reanalysis forced versus observation forced runs. Fifth, Table 3 lists the vegetation type at the four sites as alpine marsh meadow. Why is the number 3 in Figure 2?

6. Figures 3 and 4: First, what is the meaning of the horizontal error bar for each point? Second, the author attributes the cold bias of reanalysis to lead to the colder simulated MAGT and shallower ALT. However, I was wondering does the snow density gives any influence? Because $250 \, \text{kg} \, \text{m}^{-3}$ may be high for the TP, several studies (e.g., Dai et al., 2018, Yin et al., 2021) report values closer to $150 \, \text{kg} \, \text{m}^{-3}$.

7. Section 5.4: Because the text first discusses the time series of permafrost-extent anomalies, consider swapping the order of Figs. 9(b) and 9(a). Also, I could not locate the source of the 5.2% figure cited on line 289; please clarify.

8. A residual water content of $5 \, \text{m}^3 \, \text{m}^{-3}$ seems implausible in line 391. Please check the value (units/decimal).

9. Please ensure consistent verb tense throughout the manuscript, e.g., line 306: "introduce and demonstrated".

Reference:

Dai, L., Che, T., Xie, H., and Wu, X. 2018. Estimation of Snow Depth over the Qinghai-Tibetan Plateau Based on AMSR-E and MODIS Data. Remote Sensing, 10, 1989.

Yin G., Niu, F., Lin, Z., Luo, J., and Liu, M. 2021. Data-driven spatiotemporal projections of shallow permafrost based on CMIP6 across the Qinghai–Tibet Plateau at 1 $km^2$ scale. Advances in Climate Change Research, 12, 814–827.

---

## Author Comment (AC2)

**Author's Responses to RC1's comments on "Ensemble numerical simulation of permafrost over the Tibetan Plateau from Flexible Permafrost Model: 1950–2023"**

Wen Sun and Bin Cao

State Key Laboratory of Tibetan Plateau Earth System Environment and Resources (TPESER), National Tibetan Plateau Data Center (TPDC), Institute of Tibetan Plateau Research, Chinese Academy of Sciences, Beijing, China

Correspondence: Bin Cao (bin.cao@itpcas.ac.cn)

The authors would like to thank the reviewer for their constructive feedback, and the thorough assessment of the manuscript. Below, we provide a point-by-point response to each comment. Reviewer comments are given in black and responses in blue. Additionally, we have included details of how we intend to address these changes in a revised submission.

Permafrost underlies roughly 15% of the Northern Hemisphere's exposed land. Its thaw is reshaping hydrology and ecosystems and undermining the stability of infrastructure. Understanding the trajectory of permafrost dynamics under continued warming is therefore essential. At regional-to-hemispheric scales, numerical models are essential for reconstructing past states, attributing observed trends, and projecting future permafrost dynamics. Here, Sun and Cao introduce the Flexible Permafrost Model (FPM), a standalone land-surface scheme configured in one-dimensional heat conduction, and apply it to a long (1950–2023) ensemble simulation over the Tibetan Plateau (TP). The experiments are forced by ERA5-Land reanalysis data and use a deep soil column (150 m) to reconstruct the permafrost thermal regime. A 45-member ensemble to represent the broad uncertainty of hydrological parameters, and yields spatially consistent estimates of active-layer thickness (ALT), mean annual ground temperature (MAGT), and permafrost areas with observed and previous studies. Evaluation against site observations shows skill of the correct order of magnitude, and the experiments clarify how shallow-column diagnostics can bias permafrost area and trend estimates relative to deep-column simulations.

Overall, this work should be considered by The Cryosphere, provided that the authors address the comments below and supply the requested clarifications.

**Comments**

1. The author stated that the geothermal numerical model lacks the link with the atmosphere, and the land surface models are not good at representing the permafrost processes. FPM coupled the advantages of these two models to deal with the land-atmosphere interactions and extend the soil column more deeply. Please articulate the specific advantages of FPM relative to existing models: e.g., demonstrably higher accuracy, computational efficiency, or novel parameterizations that capture landscape dynamics.

Responses: RC#2 also suggested to better "explain how this new model positions itself relative to the existing ones in order to justify its relevance". In the revision, we will revise as below to clarify.

Significant efforts have been made to understand the permafrost changes over the TP based on simulations. A significant portion of these contributions comes from the hydrological community, employing models originally designed to simulate hydrological processes in permafrost-affected regions. However, many of the models implemented detailed representations of hydrological processes (e.g., water mass balance) while simplifying the surface energy balance and soil thermal processes. For instance, the DHTC model parameterizes ground heat conduction as a linear function of net radiation (Linmao et al., 2024), and the FLEXTopo-FS model uses the Stefan equation rather than a numerical solution for heat conduction (Gao et al., 2022). Beyond such hydrological models, the process-based models used for recent transient permafrost simulation over the TP can be generally divided into geothermal numerical models (i.e., GIPL model) and the common land surface models (i.e., CLM and Noah-MP). The geothermal numerical models typically have rich permafrost-specific processes, such as suitable numerical solver in heat transfer with soil phase changes (Nicolsky et al., 2007; Tubini et al., 2021), deep soil column (tens to hundreds of meters), and well-defined lower boundary, but lack representation of land-atmosphere interactions (i.e., Qin et al., 2017, Sun et al., 2023). On the other hand, the land surface models benefits from the consideration of land-atmosphere processes, and therefore outperform in describing the responses and influences of permafrost to climate warming (i.e., Guo et al., 2018, Wu et al., 2018, Zhang et al., 2021, Cao et al., 2022). Recently, a few

permafrost-specific land surface scheme models—combining the advantages of these two types of models—were proposed. The stand-alone models yield promising potential for application to cross-scale permafrost processes (Fiddes et al., 2015, Westermann et al., 2016). However, dedicated stand-alone permafrost models remain scarce for the TP. Most existing simulations rely on distributed hydrological models that have been enhanced with permafrost process representations (e.g., Gao et al., 2018; Song et al., 2020). Although these models generally offer more realistic and detailed simulations of permafrost-influenced hydrological processes, they are typically confined to site or regional scales and short time periods due to their demand for extensive spatial data and high computational cost (e.g., Pan et al., 2016; Zhang et al., 2017; Zheng et al., 2020).

2. Lines 5–6: The author states, "The FPM accounts for both vertical and lateral heat flow ...". Yet the present application appears strictly 1-D. Please make this distinction explicit in the Abstract/Introduction/Methods to avoid implying that lateral heat-flux parameterizations are active in this study, or provide details if they are.

We recognize that references to 2D capabilities could be potentially misleading (Referee #2 raised a similar point), as a full assessment of 2D-model suitability requires further applications and evaluation. In the revised manuscript, we will remove the description of lateral heat transfer in FPM from the model description section and relocate it to the outlook section.

3. Section 3.3: I was wondering do the ensemble parameters come from both Table 1 and Table 2 (Lines 177–178), or only from the hydrological parameters in Table 2? In addition, please justify the choice of 45 members per grid cell. Response: We agree this is misleading. Only the hydrological parameters in Table 2 are used to produce the ensemble member. In the revision, this part will be revised as:

"In this study, the ensemble simulation is produced using reasonable ranges of parameters (Table 2)."

4. Line 190: Could you clarify the spin-up convergence criterion? For example, which variables were evaluated, and what thresholds or tolerances were applied to judge convergence? I also suggest considering a dynamic spin-up for regional runs, which may be more efficient than a fixed 1000-year spin-up per grid cell.

Response: The soil temperature (difference for annual mean soil temperature

Figure 2: Comparison of simulated and observed day-of-year soil temperature in the active layer across the synthesis sites. The daily soil temperature present is averaged for each vegetation type and soil depth based on all available sites and years. The soil depth and numbers of sites (N) are given in parentheses. The sites used for each vegetation type and depth differ based on data availability. Observations are in black, red lines show the simulation forced by reanalyses, and the blue lines represent that forced by observed atmospheric forcing and *in situ* soil information (if available). The shaded areas depict the ensemble range from the 25th to 75th. The ensemble of observation forced simulation are produced using results from different sites and additional ranges of soil moisture (see Table 2).

Figure 3: Evaluation of modeled active layer thickness (ALT). The ensemble mean from FPM simulations (MOD-ERA5L) are given in black dot, with the whiskers representing the range between the  $25^{th}$  and  $75^{th}$  percentiles. The observed mean was aggregated from multiple measurements at a single site or from multiple sites within the same grid. N indicates the number of grids used for evaluation after aggregating sites within the same grid, and the number of measurements was given in parentheses. The additional simulation driven by observed meteorological forcing (MOD-Obs) are given in blue. Dashed lines indicate  $\pm 1$  m.

6. Figures 3 and 4: First, what is the meaning of the horizontal error bar for each point? Second, the author attributes the cold bias of reanalysis to lead to the colder simulated MAGT and shallower ALT. However, I was wondering does the snow density gives any influence? Because 250 kg m-3 may be high for the TP, several studies (e.g., Dai et al., 2018, Yin et al., 2021) report values closer to 150 kg m-3.

Response: The horizontal error bar is the range between the 25th and 75th percentiles for measured grids. This could be either from single site with multi-years' measurements or several sites in the same grid. In the revision, the figure and caption will be revised as above to clarify.

Regarding snow density, some studies (e.g., Dai et al., 2018; Yin et al., 2021) have adopted a bulk snow density of approximately 150 kg m-3 for the TP, based on observations from the China Meteorological Administration (CMA). However, recent investigations (e.g., Zhong et al., 2021; Cao et al., 2023; Che et al., 2025) suggest that the snow density values from the CMA network are significantly underestimated when compared with stand-alone measurements. This discrepancy is likely attributed to the CMA's measurement methodology, which employs a heavy snow gauge consisting of a steelyard balance and a 5000 cm3 tube-cutter, particularly problematic given the generally shallow snowpack over the TP – with a mean snow depth of only 0.01 m across 87 CMA stations. In our preprint, we used a value of 250 kg m-3, which represents a typical constant snow density.

Both RC#2 and RC#3 raised concern regarding the potential uncertainties arising from the use of a static snow density. To address this, we will incorporate the empirical snow compaction parameterization from Verseghy (1991) into the FPM. In this scheme, the fresh snow density is set to 100 kg m-3, and snow compaction is calculated as follows:

$$\rho_{sn}^{t+\Delta t} = (\rho_{sn}^t - \rho_{sn}^{max}) \cdot \exp\left(-0.24\Delta t\right) + \rho_{sn}^{max} \tag{1}$$

where  $\rho_{\rm sn}^{\rm max}$  is assumed to be 300 (kg m-3), and  $\Delta t$  is the simulation time step in day.

The updated simulations are very close to those using the static snow density of  $250 \text{ kg m}^{-3}$  as snow is very minor in most permafrost region over the TP.

Figure 4: Same as Figure 3, but for the mean annual ground temperature (MAGT). Dashed lines indicate  $\pm$  1  $^{\circ}$  C.

Figure 9: (a) Comparison of estimated current (2010–2023) permafrost area from FPM simulations with referenced estimations, and (b) anomaly of permafrost area since 1950. The subscript means the depth above which permafrost is diagnosed. The permafrost area trend ( $10^4 \text{ km}^2 \text{ dec}^{-1}$ ) is estimated for the periods before and after 1980, separately.

- 7. Section 5.4: Because the text first discusses the time series of permafrost-extent anomalies, consider swapping the order of Figs. 9(b) and 9(a). Also, I could not locate the source of the 5.2% figure cited on line 289; please clarify. Response: We will swap the order of Figs 9(b) and 9 (a) as above. The 5.2% indicated increased permafrost area between 1950–1980. In the revision, it will be updated to 2.6% based on the improved simulations with snow compaction scheme.
- 8. A residual water content of 5 m $^3$  m $^{-3}$  seems implausible in line 391. Please check the value (units/decimal). Response: It should be 0.05 m $^3$  m $^{-3}$ , will be revised.
- 9. Please ensure consistent verb tense throughout the manuscript, e.g., line 306: "introduce and demonstrated". Response: The verb tense will be revised throughout the manuscript in the revision.

**References:**

- Cao, B., Wang, S., Hao, J., Sun, W., and Zhang, K.: Inconsistency and correction of manually observed ground surface temperatures over snow-covered regions, Agricultural and Forest Meteorology, 338, 109518, 2023.
- Che, T., Dai, L., and Li, X.: Spatiotemporal distribution of seasonal snow density in the Northern Hemisphere based on in situ observation, Research in Cold and Arid Regions, 17, 137–144, https://doi.org/10.1016/j.rcar.2025.02.004, 2025.
- Dai, L., Che, T., Xie, H., and Wu, X. 2018. Estimation of Snow Depth over the QinghaiTibetan Plateau Based on AMSR-E and MODIS Data. Remote Sensing, 10, 1989.
- Yin G., Niu, F., Lin, Z., Luo, J., and Liu, M. 2021. Data-driven spatiotemporal projections of shallow permafrost based on CMIP6 across the Qinghai-Tibet Plateau at 1 km2 scale. Advances in Climate Change Research, 12, 814–827.

---

## Author Comment (AC3)

**Author's Responses to RC2's comments on "Ensemble numerical simulation of permafrost over the Tibetan Plateau from Flexible Permafrost Model: 1950–2023"**

Wen Sun and Bin Cao

State Key Laboratory of Tibetan Plateau Earth System Environment and Resources (TPESER), National Tibetan Plateau Data Center (TPDC), Institute of Tibetan Plateau Research, Chinese Academy of Sciences, Beijing, China

Correspondence: Bin Cao (bin.cao@itpcas.ac.cn)

The authors would like to thank the reviewer for their constructive feedback and thorough assessment of our manuscript. Below, we provide a point-by-point response to each comment, reviewer comments are given in black, responses are given in blue. Additionally, we have included details of how we intend to address these changes in a potential revised submission. Revised figure/table are presented at the end of our responses.

**General Comment**

In their study, Sun and Cao, present a new permafrost model, evaluate its performance at certain locations and apply it to the whole Tibetan plateau. They compare observations with simulations forced with local weather station records and with large scale reanalysis datasets. They discuss what makes the model perform better or worse and presents results on the evolution of Tibetan permafrost since the 80s. The model includes Surface Energy Balance calculation but its calculation and its coupling to the energy budget of the soil column is, from my understanding, either poorly described or problematic in its design (see my Important Comments). It resolves heat conduction in the ground with effective heat capacity, freezing curves but the water content of each cell is static (no infiltration or upward suction via evaporation and matrix potential). At the surface it also includes a snowpack module that does not consider snow mass balance.

The model claims to be flexible and to propose novel parameterizations, but according to me the flexibility is not explained or demonstrated (see my Important Point on L58) and I do not see novel parameterizations. Additionally, I do not see what are the new possibilities that this model offers compared to already existing models. I think the study needs to better present what makes it original and motivated its development. A detailed summary of its strength and weakness compared to other models would help understand the motivations for its development.

Also, in my view, the study might be more aligned with the scope of GMD than TC, given that its primary focus is on presenting a new geoscientific model. Typically, I would expect TC publications to emphasize scientific results or insights directly related to the cryosphere, whereas GMD is intended for studies of this nature. That said, since the manuscript has already passed the initial editorial screening and the editor has decided to proceed with the review process, I take it that its placement here is considered appropriate.

Altogether, for now, I have the feeling that the model in itself is not particularly novel/needed by the community (but I am happy to be proven wrong) and, unless I am mistaken, its description includes important flaws that needs to be addressed. Therefore as it is, I recommend major revisions to address those crucial points. I have not provided detailed comments on the rest of the manuscript at this stage, as I believe the major issues outlined above should be addressed before a more thorough evaluation is meaningful. Overall, I did not identify major issues with the model setup, validation, results, or discussion. However, I find the robustness of the results at the Tibetan scale questionable, given the discrepancies between observations and simulations when using the reanalyses.

Responses: We fully agree that the model will significantly benefit from implementing a better described snow and hydrology schemes as we've discussed in Sec. 6.2 Model limitations. In the revision, the snow compaction algorithm from Verseghy (1991) will be introduced to replace the static snow density (Eq. 1), and the uncertainties of the static soil moisture will be better quantified based on the ensemble spread. Below are our detailed clarifications to the concerns regarding snow density and soil moisture, along with the corresponding changes made to the possible revision.

$$\rho_{sn}^{t+\Delta t} = (\rho_{sn}^t - \rho_{sn}^{max}) \cdot \exp\left(-0.24\Delta t\right) + \rho_{sn}^{max} \tag{1}$$

where  $\rho_{sn}^{max}$  is assumed to be 300 kg m-3, and  $\Delta t$  is the simulation time step in day. The fresh snow density was set as 100

 $kg m^{-3}$ .

**Snow density**

The significant influences of snow cover on soil thermal regime have been well documented (Zhang, 2005). The required degree of model complexity depending on the intended applications. Over the Tibetan Plateau (TP), snow cover is minor, with a mean snow depth of about 1 cm (Dec–Feb) according to ground observations from a network of 87 stations (Cao et al., 2019). Consequently, the snow insulation effects are relatively minor in this region. To address the possible uncertainties using the static snow density of 250 kg m-3, additional three simulation experiments were conducted and discussed here, and the snow compaction algorithm from Verseghy (1991) will be used in the revision.

Additional three simulation experiments with different snow schemes:

- (1) static snow density of 225 kg m-3 (as -10% of 250 kg m-3);
- (2) static snow density of 275 kg m-3 (as +10% of 250 kg m-3);
- (3) the snow compaction algorithm following Verseghy (1991), with the fresh snow density of  $100 \text{ kg m}^{-3}$  and the maximum snow density of  $300 \text{ kg m}^{-3}$ .

Our simulation results indicate that:

- (1) a smaller (225 kg m-3) static snow density generally leads to a deeper ALT and warmer MAGT, but the difference is very small. The ALT difference in about 71% cells are found < 0.05 m, and the overall MAGT difference at 15 m depth was about 0.18 °C (Fig. R1a and b);
- (2) Similar to (1), a larger (275 kg m-3) static snow density generally leads to a shallower ALT and colder MAGT, but the difference is small as well (Fig. R1c and d);
- (3) the mean snow density derived from dynamic snow density scheme was about 252.9 kg m-3 during Dec–Feb, which is very close the typical value we used in preprint;
- (4) the overall difference of ALT using snow compaction algorithm (compared to the static snow density of 250 kg m $^{-3}$ ) was not remarkable with about 62% cells

Figure R1: The difference of simulated active layer thickness (ALT) and permafrost mean annual ground temperature (MAGT, 15 m) between using the static snow density of 250 kg m $^{-3}$  and 225 kg m $^{-3}$  (a, b), 275 kg m $^{-3}$  (c, d), and a empirical-based dynamic snow compaction parameterization from Verseghy (1991) (e, f). The differences derived as the simulation with static density of 250 misused by the new simulation.

Figure R2: The standard deviation of simulated active layer thickness (ALT) and mean annual ground temperature (MAGT) based on the 45-member ensemble simulations which accounted the soil moisture spread in root and vadose zones.

Figure R3: The standard deviation of the soil moisture spread in root (a) and vadose (b) zones.

Table 2: Soil moisture (m³ m⁻³) parameters selected for ensemble simulations. The dry and wet variants indicate the parameter ensemble range, and default indicates the standard choice used in model simulation.

| Soil layer | Root layer                 | Vadose layer                                         |
|------------|----------------------------|------------------------------------------------------|
| Symbol     | $\Theta_R$                 | $\theta_{ m v}$                                      |
| Default    | ensemble mean 1 | $\frac{\theta_{\text{sat}} + \theta_{\text{fc}}}{2}$ |
| Dry        | $-\mathrm{std.}^2$         | $-0.1(\theta_{\rm sat}-\overline{\theta}_{\rm fc})$  |
| Wet        | +std.                      | $+0.1(\theta_{\rm sat}-\theta_{\rm fc})$             |
| Step       | std.
4           | $0.05(\theta_{sat} - \theta_{fc})$                   |

The footnote of 1 and 2 mean the ensemble mean and standard deviation (std.) of five remote-sensing-based soil moisture in Table 1.

**TC vs. GMD**

Response: This manuscript has two primary objectives: (1) to introduce the proposed model, and (2) employ it in analyzing long-term permafrost changes over the TP. Consequently, this study extends beyond a purely methodological description, as evidenced in the Results section, which is largely devoted to presenting the spatiotemporal dynamics of permafrost. To underscore this focus, the final paragraph of the Introduction will be revised as follows to provide clarity.

"In this study, we introduce a new land surface scheme specifically designed for permafrost applications, the Flexible Permafrost Model (FPM). This model serves as a flexible platform for a variety of permafrost processes. The suitability of the new model was carefully evaluated, we then employed it in analyzing the long-term (1950–2023) permafrost thermal regime over the TP based on the ensemble simulation. Specially, this study"

- 1. gives a detailed description of the model conceptualization, structure, and parameterization;
- 2. evaluates the model performance in reproducing permafrost characteristics based on the ensemble approach, such as active layer thickness (ALT), and the thermal state;
- 3. interprets current conditions and historical changes of permafrost in response to climate change from the stand-alone simulations;
- 4. proposes insights for future model developments.

**Important comments**

**L30-43**

When a new model is published, it becomes part of the existing landscape of models, and it is important to explain how this new model positions itself relative to the existing ones in order to justify its relevance. I think the study should be

more thorough in this regard and more exhaustive regarding which permafrost models are used in Tibet to study what. The Introduction should be expanded in consequence (see my comments about L42 and L43). Response: We agree. We further reviewed current physical-based models over the TP.

Significant efforts have been made to understand the permafrost changes over the TP based on simulations. A significant portion of these contributions comes from the hydrological community, employing models originally designed to simulate hydrological processes in permafrost-affected regions. However, many of the models implemented detailed representations of hydrological processes (e.g., water mass balance) while simplifying the surface energy balance and soil thermal processes. For instance, the DHTC model parameterizes ground heat conduction as a linear function of net radiation (Linmao et al., 2024), and the FLEXTopo-FS model uses the Stefan equation rather than a numerical solution for heat conduction (Gao et al., 2022). Beyond such hydrological models, the process-based models used for recent transient permafrost simulation over the TP can be generally divided into geothermal numerical models (i.e., GIPL model) and the common land surface models (i.e., CLM and Noah-MP). The geothermal numerical models typically have rich permafrost-specific processes, such as suitable numerical solver in heat transfer with soil phase changes (Nicolsky et al., 2007; Tubini et al., 2021), deep soil column (tens to hundreds of meters), and well-defined lower boundary, but lack representation of landatmosphere interactions (i.e., Qin et al., 2017, Sun et al., 2023). On the other hand, the land surface models benefits from the consideration of land-atmosphere processes, and therefore outperform in describing the responses and influences of permafrost to climate warming (i.e., Guo et al., 2018, Wu et al., 2018, Zhang et al., 2021, Cao et al., 2022). Recently, a few permafrost-specific land surface scheme models-combining the advantages of these two types of models-were proposed. The stand-alone models yield promising potential for application to cross-scale permafrost processes (Fiddes et al., 2015, Westermann et al., 2016). However, dedicated stand-alone permafrost models remain scarce for the TP. Most existing simulations rely on distributed hydrological models that have been enhanced with permafrost process representations (e.g., Gao et al., 2018; Song et al., 2020). Although these models generally offer more realistic and detailed simulations of permafrost-influenced hydrological processes, they are typically confined to site or regional scales and short time periods due to their demand for extensive spatial data and high computational cost (e.g., Pan et al., 2016; Zhang et al., 2017; Zheng et al., 2020).

**L58**

"The application of FPM with lateral heat is provided in Sun et al. (2023)." This is very confusing to me for two reasons. First, vertical 1D models and cross sectional 2D models are usually very different types of models that implies different formulation of their physical equations, different numerical schemes for their resolution and different types of upper boundary conditions. The present study mentions surface energy balance calculation whereas Sun et al. (2023) forced their model with ground surface temperatures. So I do not understand how a 1D model can become a full 2D model (not a coupling of 1D simulation together).

Second, the given explanation is confusing. Sun et al. (2023) says "A 2D heat conduction model developed by Ling and Zhang (2004a) was used to simulate the permafrost thermal regime." So if it is the same model in 1D and 2D, is the present study actually presenting a new model or is it presenting improvements brought to a model published initially in 2004? Since the goal of this study is to present a new model, this kind of question need to clarified to consider publication. Responses: We fully agree that 2D models are fundamentally different from 1D models. However, it is possible for a single model to support implementations in 1D, 2D, and even 3D. An example from the permafrost modeling community is the Control Volume Permafrost Model (CVPM) by Clow (2018), which "implements the nonlinear heat-transfer equations in 1-D, 2-D, and 3-D Cartesian coordinates, as well as in 1-D radial and 2-D cylindrical coordinates.".

We recognize that references to 2D capabilities could be potentially misleading (Referee #1 raised a similar point), as a full assessment of 2D-model suitability requires further applications and evaluation. In the revised manuscript, we will remove the description of lateral heat transfer in FPM from the model description section and relocate it to the outlook section.

**L62-104**

"A physically-based surface energy balance scheme for different land surface cover types with varying snow regimes and properties was coupled to FPM, and was formulated as:

$$(1-\alpha)Qsi + Qli + Qle + Qh + Qe + Qc = Qm$$
"

Major problem here. The study intends to describe the ground surface SEB (because we are in part 2.1 of a study presenting a permafrost model, with no mention to the snow scheme yet). Yet it looks like a description of a snow SEB, as evidenced by the expression "varying snow regime" and by the fact that the equation calculates energy for melt, which is inappropriate for a ground surface scheme. For a ground surface scheme, we want the SEB to give us access to the dE/dt of the surface so that it can force the heat diffusion/advection in the soil column. If we do not have that, we do not have the coupling between the climate and the temperature in the ground. This problem persist over the whole section 2.1. These are very important aspect of the model description that need to be carefully addressed so that we can understand what we are talking about. Response: We agree that the term  $Q_m$  is not appropriate here, as snow mass balance is not implemented in the current

version of FPM. In the code,  $Q_m$  is set to zero and therefore does not affect simulation results. In the revision,  $Q_m$  will be removed.

Although both temperature and heat flux can serve as the upper boundary condition for soil heat conduction, FPM uses the surface temperature ( $T_{s0}$ )—whether snow or soil—similar to the approach in the CryoGrid community model (e.g., Eqs. 4 and 5 in Westermann et al., 2023). Accordingly, the surface energy balance will be revised as:

$$(1-\alpha)Qsi + Qli + Qle + Qh + Qe = Qc$$
(2)

and the sign of Qc should be revised,

$$Q_c = (T_{s0} - T_g)(\frac{z_{sn}}{k_{sn}} + \frac{z_g}{k_g})^{-1}$$
(3)

In FPM,  $Q_{le}$  (Eq. 3) and  $Q_{h}$  (Eq. 4) are functions of  $T_{s0}$ .  $Q_{e}$  is derived as a function of  $Q_{n}$  and  $Q_{c}$  and thus also depends on  $T_{s0}$ . For each time step,  $T_{s0}$ , the upper boundary condition for the subsurface, is solved iteratively using the Newton-Raphson method to ensure energy conservation.

L78 and 86

I have 2 problems with this quantification of the turbulent fluxes. First, if you take Preistley and Taylor for  $Q_e$ , then you have to take  $Q_h$  as the residual of the available energy for turbulent fluxes, which is:

$$Q_h = (1 - \alpha \Delta / (\Delta + \gamma)) \times (Q_n - Q_g)$$

Otherwise your SEB is not energy conservative. Preistley and Taylor considers that the energy available for both turbulent fluxes is  $Q_n - Q_g$  and provide a formula to find what fraction of that available energy goes to the latent flux. So, in order to have a consistent surface energy balance budget,  $Q_h$  has to be the complementary fraction of  $Q_n - Q_g$ , not another formula based on another theory.

Response: We agree that using a consistent scheme for  $Q_h$  and  $Q_e$  would enhance model robustness. That said, combined approaches have been employed in previous studies. For instance, Song et al. used both Priestley-Taylor (for  $Q_e$ ) and Monin-Obukhov similarity theory (for  $Q_h$ ) in their surface energy balance model; other examples include Agam et al. (2010) and Kustas et al. (2003). Our detailed evaluations at both site and regional scales have demonstrated the suitability of FPM. We acknowledge that using two different theories may introduce additional uncertainties, and we will address this issue in the Discussion section in the revised manuscript.

Second, if you use Preistley and Taylor you will assume that the sum of the terms of your SEB equals 0 and not the energy variation of the soil surface. Otherwise you'd have to work with a modified version that would look like:

$$Q_e = \alpha \Delta / (\Delta + \gamma)(Q_n - Q_g - (\partial E_{surf})/\partial t)$$

But then you would not be able to calculate together the turbulent flux and the energy variation of the surface (one too many unknown). Yet, it is through the energy variation of the surface that you can couple the SEB and the energy budget at depth in the ground (because the energy budget of the surface will drive the one of the subsurface). Therefore, this whole SEB description is very confusing to me and absolutely need to be fixed. For now I cannot understand the coupling with the subsurface

Response: We agree that the current presentation of the SEB is misleading. As clarified above, energy conservation in the SEB is achieved through Eq. (1) and numerical solution via the Newton-Raphson iterative method.

**Specific comments**

L14: "shallow soil columns" indicate typical depth

Response: Will be revised as below

"Furthermore, our findings suggest that current land surface models, which utilize shallow soil columns (typically  $\sim 3$  m)"

L36: Wrong reference with Lan et al. 2025 here? It is a review of the strength and weakness of a climate reanalysis datasets, it does not present a permafrost model

Response: Yes, it is a reanalysis evaluation paper. Lan et al., 2025 indicated the numerical solution, i.e., decoupled energy conservation parameterization (DECP), used in many land surface models may be an issue for permafrost simulations. To clarify, the reference will be replaced by two more related references, i.e., Nicolsky et al., 2007; Tubini et al., 2021. This part will be revised as below.

"The geothermal numerical models typically have rich permafrost-specific processes, such as suitable numerical solver in heat transfer with soil phase changes (Nicolsky et al., 2007; Tubini et al., 2021), deep soil column..."

L43: I am surprised the authors do not mention other models used to study Tibetan permafrost like the GBEHM model that is greatly used within the Chinese community (Fang et al., 2025; Gao et al., 2018; Qin et al., 2017; Shi et al., 2020; Wang et al., 2023, 2018; Wang and Gao, 2022, 2025; Yang et al., 2023b, a), or the recent DHTC model (Linmao et al., 2024) and FLEXTopo-FS model (Gao et al., 2022).

Response: We agree that some hydrological models applied in permafrost regions were mentioned in this section. This is primarily because such models typically include more detailed hydrological processes (e.g., water mass balance) but simplify the surface energy balance and heat conduction processes. For example, the DHTC model (Linmao et al., 2024) parameterizes ground heat conduction as a linear function of net radiation, and the FLEXTopo-FS model (Gao et al., 2022) uses the Stefan equation rather than a numerical solution for heat conduction. In the revision, we will reformulate the this paragraph to clarify (see below).

Actually, Qin et al. (2017) used GIPL, not GBEHM. Nonetheless, both GBEHM (cited in L43 as Zheng et al., 2020) and GIPL (cited in L37 as Qin et al., 2017) are referenced in the preprint.

Significant efforts have been made to understand the permafrost changes over the TP based on simulations. A significant portion of these contributions comes from the hydrological community, employing models originally designed to simulate hydrological processes in permafrost-affected regions. However, many of the models implemented detailed representations of hydrological processes (e.g., water mass balance) while simplifying the surface energy balance and soil thermal processes. For instance, the DHTC model parameterizes ground heat conduction as a linear function of net radiation (Linmao et al., 2024), and the FLEXTopo-FS model uses the Stefan equation rather than a numerical solution for heat conduction (Gao et al., 2022). Beyond such hydrological models, the process-based models used for recent transient permafrost simulation over the TP can be generally divided into geothermal numerical models (i.e., GIPL model) and the common land surface models (i.e., CLM and Noah-MP). The geothermal numerical models typically have rich permafrost-specific processes, such as suitable numerical solver in heat transfer with soil phase changes (Nicolsky et al., 2007; Tubini et al., 2021), deep soil column (tens to hundreds of meters), and well-defined lower boundary, but lack representation of landatmosphere interactions (i.e., Qin et al., 2017, Sun et al., 2023). On the other hand, the land surface models benefits from the consideration of land-atmosphere processes, and therefore outperform in describing the responses and influences of permafrost to climate warming (i.e., Guo et al., 2018, Wu et al., 2018, Zhang et al., 2021, Cao et al., 2022). Recently, a few permafrost-specific land surface scheme models-combining the advantages of these two types of models-were proposed. The stand-alone models yield promising potential for application to cross-scale permafrost processes (Fiddes et al., 2015, Westermann et al., 2016). However, dedicated stand-alone permafrost models remain scarce for the TP. Most existing simulations rely on distributed hydrological models that have been enhanced with permafrost process representations (e.g., Gao et al., 2018; Song et al., 2020). Although these models generally offer more realistic and detailed simulations of permafrostinfluenced hydrological processes, they are typically confined to site or regional scales and short time periods due to their demand for extensive spatial data and high computational cost (e.g., Pan et al., 2016; Zhang et al., 2017; Zheng et al., 2020).

Eq13: I would avoid writing the equation, it gives the impression that your are describing how the model works during simulations whereas you just used the equation once for the evaluation of static parameters. I would rather just state the values of k and C in the text and say that they were calculated based on the ref you mention.

Response: In the revision, empirical-based snow compaction parameterization from Verseghy (1991) will be introduced to FPM, and, therefore we decided to keep it.

Fig.2: "Comparison of simulated active layer soil temperature with time series at the synthesis sites." What is the methodology here? The active layer can be pretty deep. You averaged the temperature over the whole the active layer at a daily time step? Also what do the red and blue numbers with and without parenthesis correspond to? I assume from the text that it is the bias? It should be written in the caption for more clarity.

Response: The figure and caption will be revised as below to clarify.

Fig.3: The methodology described in the caption is hard to understand, please elaborate more.

Response: The figure and caption will be revised as below to clarify.

**Phrasing and typos**

L214: Align

Response: will be revised.

L218: "via the processes of latent heat and soil moisture" the reader understands, but these are not processes, please rephrase. Response: This sentence will be changed as below to clarify.

"FPM considers the influences of vegetation on permafrost via the latent heat exchange and soil moisture effects. (Appendix A)."

L221: "The remote-sensing datasets are different in temporal coverage, so we use the climatology to represent the long-term conditions." I think the phrasing "use the climatology" can be improved.

Response: This sentence will be changed as below in the next revision.

"FPM considers the influences of vegetation on permafrost via the latent heat and soil moisture etc. (Appendix A). In FPM, static vegetation is assumed and the vegetation optical depth (VOD), leaf area index (LAI), and vegetation type are required (Table 1). For snow-free periods, the ground albedo is from Jia et al. (2022).

The remote-sensing datasets vary in their temporal coverage, so we used the climatology to represent the long-term conditions. For the VOD and snow-free ground albedo, the daily measurements over the entire recording period were aggregated into a day-of-year climatology using the median, so as to reduce sensitivity to extreme values. The monthly LAI from Myneni et al. (2021) was aggregated to monthly medians. Daily  $\theta_R$  values were first aggregated into monthly averages for each dataset. These monthly values from the thawing season (June to August) were then used to compute the annual mean. For each soil moisture dataset, the average over the entire recording period was derived, and an ensemble mean across the five datasets was calculated and employed as model inputs. Note that only the measurements from the thawing season (June to August) were used to derive VOD and  $\theta_R$ ."

Fig.2: "The daily soil temperature present are averaged", syntax problem. Response: the caption will be revised as below to clarify.

"The daily soil temperature data were averaged across all available sites and years for each vegetation type at different soil depths."

Figure 2: Comparison of simulated and observed day-of-year soil temperature in the active layer across the synthesis sites. The daily soil temperature present is averaged for each vegetation type and soil depth based on all available sites and years. The soil depth and numbers of sites (N) are given in parentheses. The sites used for each vegetation type and depth differ based on data availability. Observations are in black, red lines show the simulation forced by reanalyses, and the blue lines represent that forced by observed atmospheric forcing and *in situ* soil information (if available). The shaded areas depict the ensemble range from the 25th to 75th. The ensemble of observation forced simulation are produced using results from different sites and additional ranges of soil moisture (see Table 2).

Figure 3: Evaluation of modeled active layer thickness (ALT). The ensemble mean from FPM simulations (MOD-ERA5L) are given in black dot, with the whiskers representing the range between the  $25^{th}$  and  $75^{th}$  percentiles. The observed mean was aggregated from multiple measurements at a single site or from multiple sites within the same grid. N indicates the number of grids used for evaluation after aggregating sites within the same grid, and the number of measurements was given in parentheses. The additional simulation driven by observed meteorological forcing (MOD-Obs) are given in blue. Dashed lines indicate  $\pm$  1 m.

**References:**

- Agam, N., Kustas, W. P., Anderson, M. C., Norman, J. M., Colaizzi, P. D., Howell, T. A., Prueger, J. H., Meyers, T. P., and Wilson, T. B. (2010): Application of the Priestley-Taylor Approach in a Two-Source Surface Energy Balance Model, Journal of Hydrometeorology, 11, 185–198, https://doi.org/10.1175/2009jhm1124.1.
- Cao, B., Zhang, T., Wu, Q., Sheng, Y., Zhao, L., and Zou, D.: Permafrost zonation index map and statistics over the Qinghai—Tibet Plateau based on field evidence, Permafrost & Periglacial, 30, 178–194, https://doi.org/10.1002/ppp.2006, 2019.
- Cao, B., Arduini, G., and Zsoter, E.: Brief communication: Improving ERA5-Land soil temperature in permafrost regions using an optimized multi-layer snow scheme, The Cryosphere, 16, 2701–2708, https://doi.org/10.5194/tc-16-2701-2022, 2022.
- Clow, G. D. (2018): CVPM 1.1: a flexible heat-transfer modeling system for permafrost, Geoscientific Model Development, 11, 4889–4908, https://doi.org/10.5194/gmd-11-4889-2018.
- Essery, R., Morin, S., Lejeune, Y., and B Mënard, C.: A comparison of 1701 snow models using observations from an alpine site, Advances in Water Resources, 55, 131–148, 2013.
- Fiddes, J. and Gruber, S.: TopoSCALE v.1.0: downscaling gridded climate data in complex terrain, Geoscientific Model Development, 7, 387–405, https://doi.org/10.5194/gmd-7-387-2014, 2014.
- Gao, B., Yang, D., Qin, Y., Wang, Y., Li, H., Zhang, Y., and Zhang, T.: Change in frozen soils and its effect on regional hydrology, upper Heihe basin, northeastern Qinghai–Tibetan Plateau, The Cryosphere, 12, 657–673, https://doi.org/10.5194/tc-12-657-2018, 2018.
- Gao, H., Han, C., Chen, R., Feng, Z., Wang, K., Fenicia, F., and Savenije, H. (2022): Frozen soil hydrological modeling for a mountainous catchment northeast of the Qinghai-Tibet Plateau, Hydrology and Earth System Sciences, 26, 4187–4208.
- Göckede, M., Kittler, F., Kwon, M. J., Burjack, I., Heimann, M., Kolle, O., Zimov, N., and Zimov, S.: Shifted energy fluxes, increased Bowen ratios, and reduced thaw depths linked with drainage-induced changes in permafrost ecosystem structure, The Cryosphere, 11, 2975–2996, https://doi.org/10.5194/tc-11-2975-2017, 2017.
- Groenke, B., Langer, M., Nitzbon, J., Westermann, S., Gallego, G., and Boike, J.: Investigating the thermal state of permafrost with Bayesian inverse modeling of heat transfer, The Cryosphere, 17, 3505–3533, https://doi.org/10.5194/tc-17-3505-2023, 2023.
- Guo, D., Wang, A., Li, D., and Hua, W.: Simulation of changes in the near-surface soil freeze/thaw cycle using clm4.5 with four atmospheric forcing data sets, Journal of Geophysical Research: Atmospheres, 123, 2509–2523, 2018.
- Ling, F. and Zhang, T.: A numerical model for surface energy balance and thermal regime of the active layer and permafrost containing unfrozen water, Cold Regions Science Technology, 38, 1–15, https://doi.org/10.1016/S0165-232X(03)00057-0, 2004.
- Linmao, G., Genxu, W., Chunlin, S., Shouqin, S., Kai, L., Jinlong, L., Yang, L., Biying, Z., Jiapei, M., and Peng, H.: Development of a modular distributed hydro-thermal coupled hydrological model for cold regions, Journal of Hydrology, 644, 132099, https://doi.org/10.1016/j.jhydrol.2024.132099, 2024.
- Ling, F., and T. Zhang (2004), Modeling study of talik freeze-up and permafrost response under drained thaw lakes on the Alaskan Arctic Coastal Plain, Journal of Geophysical Research: Atmospheres, 109, D01111.
- Myhra, K. S., Westermann, S., and Etzelmüller, B. (2017) Modelled Distribution and Temporal Evolution of Permafrost in Steep Rock Walls Along a Latitudinal Transect in Norway by CryoGrid 2D. Permafrost and Periglac. Process., 28: 172–182.
- Nicolsky, D. J., Romanovsky, V. E., Alexeev, V. A., and Lawrence, D. M.: Improved modeling of permafrost dynamics in a GCM land-surface scheme, Geophysical Research Letters, 34, https://doi.org/10.1029/2007gl029525, 2007.
- Kustas, W. P., Bindlish, R., French, A. N., and Schmugge, T. J.: Comparison of energy balance modeling schemes using microwave-derived soil moisture and radiometric surface temperature, Water Resources Research, 39, 2003.
- Orsolini, Y., Wegmann, M., Dutra, E., Liu, B., Balsamo, G., Yang, K., De Rosnay, P., Zhu, C., Wang, W., Senan, R., and Arduini, G.: Evaluation of snow depth and snow cover over the Tibetan Plateau in global reanalyses using in situ and satellite remote sensing observations, The Cryosphere, 13, 2221–2239, https://doi.org/10.5194/tc-13-2221-2019, 2019.
- Pan, X., Li, Y., Yu, Q., Shi, X., Yang, D., and Roth, K.: Effects of stratified active layers on high-altitude permafrost warming: a case study on the Qinghai-Tibet Plateau, The Cryosphere, 10, 1591–1603, 2016.
- Qin, Y., Wu, T., Zhao, L., Wu, X., Li, R., Xie, C., Pang, Q., Hu, G., Qiao, Y., Zhao, G., et al.: Numerical modeling of the active layer thickness and permafrost thermal state across Qinghai-Tibetan Plateau, Journal of Geophysical Research: Atmospheres, 122, 11-604, https://doi.org/10.1002/2017JD026858, 2017.
- Song, L., Kustas, W. P., Liu, S., Colaizzi, P. D., Nieto, H., Xu, Z., Ma, Y., Li, M., Xu, T., Agam, N., Tolk, J. A., and Evett, S. R. (2016): Applications of a thermal-based two-source energy balance model using Priestley-Taylor approach for surface temperature partitioning under advective conditions, Journal of Hydrology, 540, 574–587.
- Sun, Z., Zhao, L., Hu, G., Zhou, H., Liu, S., Qiao, Y., et al. (2022). Numerical simulation of thaw settlement and permafrost changes at three sites along the Qinghai-Tibet Engineering Corridor in a warming climate. Geophysical Research Letters, 49, e2021GL097334. https://doi.org/10.1029/2021GL097334
- Sun, W., Cao, B., Hao, J., Wang, S., Clow, G. D., Sun, Y., Fan, C., Zhao, W., Peng, X., Yao, Y., et al.: Two-dimensional simulation of island permafrost degradation in Northeastern Tibetan Plateau, Geoderma, 430, 116 330, 2023.
- Sun Z, Zhao L, Hu G, et al. Modeling permafrost changes on the Qinghai–Tibetan plateau from 1966 to 2100: A case study from two boreholes along the Qinghai–Tibet engineering corridor. Permafrost and Periglac Process. 2020; 31: 156–171. https://doi.org/10.1002/ppp.2022

- Song, L., Wang, L., Li, X., Zhou, J., Luo, D., Jin, H., Qi, J., Zeng, T., and Yin, Y.: Improving Permafrost Physics in a Distributed Cryosphere-Hydrology Model and Its Evaluations at the Upper Yellow River Basin, JGR Atmospheres, 125, e2020JD032916, https://doi.org/10.1029/2020JD032916, 2020.
- Tubini, N., Gruber, S., and Rigon, R.: A method for solving heat transfer with phase change in ice or soil that allows for large time steps while guaranteeing energy conservation, The Cryosphere, 15, 2541–2568, https://doi.org/10.5194/tc-15-2541-2021, 2021.
- Verseghy, D. L.: Class–A Canadian land surface scheme for GCMS. I. Soil model, Intl Journal of Climatology, 11, 111–133, https://doi.org/10.1002/joc.3370110202, 1991.
- Westermann, S., Langer, M., Boike, J., Heikenfeld, M., Peter, M., Etzelmüller, B., and Krinner, G.: Simulating the thermal regime and thaw processes of ice-rich permafrost ground with the land-surface model CryoGrid 3, Geoscientific Model Development, 9, 523–546, https://doi.org/10.5194/gmd-9-523-2016, 2016.
- Wu, X., Nan, Z., Zhao, S., Zhao, L., Cheng, G. J. P., and Processes, P.: Spatial modeling of permafrost distribution and properties on the Qinghai-Tibet Plateau, Permafrost Periglacial Processes, 29, 86–99, https://doi.org/10.1002/ppp.1971, 2018
- Zhang, G., Nan, Z., Yin, Z., and Zhao, L.: Isolating the Contributions of Seasonal Climate Warming to Permafrost Thermal Responses Over the Qinghai-Tibet Plateau, JGR Atmospheres, 126, e2021JD035 218, https://doi.org/10.1029/2021JD035218, 2021.
- Zhang, Y., Cheng, G., Li, X., Jin, H., Yang, D., Flerchinger, G. N., Chang, X., Bense, V. F., Han, X., and Liang, J.: Influences of Frozen Ground and Climate Change on Hydrological Processes in an Alpine Watershed: A Case Study in the Upstream Area of the Hei'he River, Northwest China: Influences of Frozen Soil Dynamics on Hydrological Processes, Permafrost and Periglac. Process., 28, 420–432, https://doi.org/10.1002/ppp.1928, 2017.
- Zhang, T.: Influence of the seasonal snow cover on the ground thermal regime: An overview, Reviews of Geophysics, 43, 2004RG000157, https://doi.org/10.1029/2004RG000157, 2005.
- Zhang, Y., Cheng, G., Li, X., Jin, H., Yang, D., Flerchinger, G. N., Chang, X., Bense, V. F., Han, X., and Liang, J.: Influences of Frozen Ground and Climate Change on Hydrological Processes in an Alpine Watershed: A Case Study in the Upstream Area of the Hei'he River, Northwest China: Influences of Frozen Soil Dynamics on Hydrological Processes, Permafrost and Periglac. Process., 28, 420–432, https://doi.org/10.1002/ppp.1928, 2017.
- Zwieback, S., Westermann, S., Langer, M., Boike, J., Marsh, P., and Berg, A.: Improving Permafrost Modeling by Assimilating Remotely Sensed Soil Moisture, Water Resources Research, 55, 1814–1832, https://doi.org/10.1029/2018wr023247, 2019.

---

## Author Comment (AC4)

**Author's Responses to RC3's comments on "Ensemble numerical simulation of permafrost over the Tibetan Plateau from Flexible Permafrost Model: 1950–2023"**

Wen Sun and Bin Cao

State Key Laboratory of Tibetan Plateau Earth System Environment and Resources (TPESER), National Tibetan Plateau Data Center (TPDC), Institute of Tibetan Plateau Research, Chinese Academy of Sciences, Beijing, China

Correspondence: Bin Cao (bin.cao@itpcas.ac.cn)

The authors would like to thank the reviewer for their constructive feedback and thorough assessment of our manuscript. Below, we provide a point-by-point response to each comment, reviewer comments are given in black, responses are given in blue. Additionally, we have included details of how we intend to address these changes in a potential revised submission. Revised figure/table are presented at the end of our responses.

The manuscript presents a new simulation framework for large scale numerical simulation of permafrost dynamics, and apply it to the quantification of permafrost metrics (ALT, MAGT, extent) of the permafrost cover over the Tibetan Plateau from 1950 to 2023. The presented simulations are suffering of strong assumptions regarding soil water content and snow cover (both 'static'), which in my opinion hampers the possibility of temporal evolution analysis. The bibliography of the permafrost modelling landscape is also incomplete.

I do not think that the manuscript may be published in TC in its present form. A significant work for better discussing the limitations of the simulations and put them in the context of permafrost modelling across scales is needed. Thus I recommend a major revision prior to reconsider whether or not it may be published in TC.

Responses: We fully agree that the model will significantly benefit from implementing a better described snow and hydrology schemes as we've discussed in Sec. 6.2 Model limitations. In the revision, the snow compaction algorithm from Verseghy (1991) will be introduced to replace the static snow density (Eq. 1), and the uncertainties of the static soil moisture will be better quantified based on the ensemble spread. Below are our detailed clarifications to the concerns regarding snow density and soil moisture, along with the corresponding changes made to the possible revision.

$$\rho_{sn}^{t+\Delta t} = (\rho_{sn}^t - \rho_{sn}^{max}) \cdot \exp(-0.24\Delta t) + \rho_{sn}^{max}$$
(1)

where  $\rho_{sn}^{max}$  is assumed to be 300 kg m-3, and  $\Delta t$  is the simulation time step in day. The fresh snow density was set as 100 kg m-3.

**Snow density**

The significant influences of snow cover on soil thermal regime have been well documented (Zhang, 2005). The required degree of model complexity depending on the intended applications. Over the Tibetan Plateau (TP), snow cover is minor, with a mean snow depth of about 1 cm (Dec–Feb) according to ground observations from a network of 87 stations (Cao et al., 2019). Consequently, the snow insulation effects are relatively minor in this region. To address the possible uncertainties using the static snow density of 250 kg m-3, additional three simulation experiments were conducted and discussed here, and the snow compaction algorithm from Verseghy (1991) will be used in the revision.

Additional three simulation experiments with different snow schemes:

- (1) static snow density of 225 kg m-3 (as -10% of 250 kg m-3);
- (2) static snow density of 275 kg  $m^{-3}$  (as +10% of 250 kg  $m^{-3}$ );
- (3) the snow compaction algorithm following Verseghy (1991), with the fresh snow density of  $100 \text{ kg m}^{-3}$  and the maximum snow density of  $300 \text{ kg m}^{-3}$ .

**Our simulation results indicate that:**

(1) a smaller (225 kg m $^{-3}$ ) static snow density generally leads to a deeper ALT and warmer MAGT, but the difference is very small. The ALT difference in about 71% cells are found < 0.05 m, and the overall MAGT difference at 15 m depth was about 0.18 °C (Fig. R1a and b);

- (2) Similar to (1), a larger (275 kg m-3) static snow density generally leads to a shallower ALT and colder MAGT, but the difference is small as well (Fig. R1c and d);
- (3) the mean snow density derived from dynamic snow density scheme was about 252.9 kg m-3 during Dec–Feb, which is very close the typical value we used in preprint;
- (4) the overall difference of ALT using snow compaction algorithm (compared to the static snow density of 250 kg m-3) was not remarkable with about 62% cells

Figure R1: The difference of simulated active layer thickness (ALT) and permafrost mean annual ground temperature (MAGT, 15 m) between using the static snow density of 250 kg m $^{-3}$  and 225 kg m $^{-3}$  (a, b), 275 kg m $^{-3}$  (c, d), and a empirical-based dynamic snow compaction parameterization from Verseghy (1991) (e, f). The differences derived as the simulation with static density of 250 misused by the new simulation.

The footnote of 1 and 2 mean the ensemble mean and standard deviation (std.) of five remote-sensing-based soil moisture in Table 1.

Table 2: Soil moisture (m³ m⁻³) parameters selected for ensemble simulations. The dry and wet variants indicate the parameter ensemble range, and default indicates the standard choice used in model simulation.

| Soil layer | Root layer                 | Vadose layer                                         |
|------------|----------------------------|------------------------------------------------------|
| Symbol     | $\theta_R$                 | $\Theta_{v}$                                         |
| Default    | ensemble mean 1 | $\frac{\theta_{\text{sat}} + \theta_{\text{fc}}}{2}$ |
| Dry        | $-std.^2$                  | $-0.1(\theta_{sat}-\tilde{\theta}_{fc})$             |
| Wet        | +std.                      | $+0.1(\theta_{\rm sat}-\theta_{\rm fc})$             |
| Step       | std.
4           | $0.05(\theta_{sat}-\theta_{fc})$                     |

Figure R2: The standard deviation of simulated active layer thickness (ALT) and mean annual ground temperature (MAGT) based on the 45-member ensemble simulations which accounted the soil moisture spread in root and vadose zones.

Figure R3: The standard deviation of the soil moisture spread in root (a) and vadose (b) zones.

L1: "Permafrost remains a largely subsurface phenomenon" Clumsy. Permafrost is a subsurface phenomenon. I guess the authors want to point out the difficulty of direct observation of this subsurface phenomena as the reason why its understanding largely relies on numerical simulations. First sentence to be rephrased.

Responses: Yes, details are given in Sec. Introduction (L26–30: Despite permafrost's importance, direct permafrost measurements, such as borehole temperature, are rare due to harsh environments and high costs (Biskaborn et al., 2015). This is especially true on the Tibetan Plateau (TP), where complex terrain and high altitudes impose further constraints on permafrost research (Cao et al., 2017b, 2019b)... Therefore, process-based simulation is an increasingly important tool for transient assessment of permafrost conditions and dynamics.).

In the revision, the sentence will be changes as

"Permafrost is a subsurface phenomenon that is difficult to be measured directly, and understanding its dynamics as well as influences under a warming climate depends critically on numerical simulations.".

L31-43: An important part of the permafrost modelling landscape is overlooked in the bibliographical survey given in the introduction: the cryohydrogeological simulators (e.g.: Grenier et al., 2018, Hu et al., 2023). These mechanistic models are based on the numerical resolution of the equations on the continuum mechanics, and thus have a much bigger predictive potential than conceptual, calibrated models. I think that, for the sake of completeness, this type of model should also be included in the survey.

Responses: We agree the cryohydrogeological simulators is not involved here. Given the numerical resolution (both the temporal and spatial ones), this kind of the cryohydrogeological simulators with more realistic and therefore complex processes are generally applied in very fine-scale (meters to several hundreds meters) studies based on very small simulation step (seconds) as given in Grenier et al., (2018) and McKenzie et al., (2007). This is because such simulations are data-intensive, computational costs and require additional boundary conditions. In other words, cryohydrogeological simulators may be challenging to be applied for the large-scale simulations as presented in this study.

On the other hand, Referee #2 suggested to review the hydrological models. In the revision, the following part will be added to clarify.

"Significant efforts have been made to understand the permafrost changes over the TP based on simulations. A significant portion of these contributions comes from the hydrological community, employing models originally designed to simulate hydrological processes in permafrost-affected regions. However, many of the models implemented detailed representations of hydrological processes (e.g., water mass balance) while simplifying the surface energy balance and soil thermal processes. For instance, the DHTC model (Linmao et al., 2024) parameterizes ground heat conduction as a linear function of net radiation, and the FLEXTopo-FS model (Gao et al., 2022) uses the Stefan equation rather than a numerical solution for heat conduction. Beyond such hydrological models, the process-based models used for recent transient permafrost simulation over the TP can be generally divided into geothermal numerical models (i.e., GIPL model) and the common land surface models (i.e., CLM and Noah-MP). The geothermal numerical models typically have rich permafrost-specific processes, such as suitable numerical solver in heat transfer with soil phase changes (Nicolsky et al., 2007; Tubini et al., 2021), deep soil column (tens to hundreds of meters), and well-defined lower boundary, but lack representation of landatmosphere interactions (i.e., Qin et al., 2017, Sun et al., 2023). On the other hand, the land surface models benefits from the consideration of land-atmosphere processes, and therefore outperform in describing the responses and influences of permafrost to climate warming (i.e., Guo et al., 2018, Wu et al., 2018, Zhang et al., 2021, Cao et al., 2022). Recently, a few permafrost-specific land surface scheme models-combining the advantages of these two types of models-were proposed. The stand-alone models yield promising potential for application to cross-scale permafrost processes (Fiddes et al., 2015, Westermann et al., 2016). However, dedicated stand-alone permafrost models remain scarce for the TP. Most existing simulations rely on distributed hydrological models that have been enhanced with permafrost process representations (e.g., Gao et al., 2018; Song et al., 2020). Although these models generally offer more realistic and detailed simulations of permafrost-influenced hydrological processes, they are typically confined to site or regional scales and short time periods due to their demand for extensive spatial data and high computational cost (e.g., Pan et al., 2016; Zhang et al., 2017; Zheng et al., 2020)."

L36: Lan et al., 2025 seems to be a reference related to a reanalysis, not to a model. Reanalysis are built using models, but they are not models.

Responses: Yes, it is a reanalysis evaluation paper. Lan et al., 2025 indicated the numerical solution, i.e., decoupled energy conservation parameterization (DECP), used in many land surface models may be an issue for permafrost simulations. To clarify, the reference will be replaced by two more related references, i.e., Nicolsky et al., 2007; Tubini et al., 2021. This part will be revised as below.

"The geothermal numerical models typically have rich permafrost-specific processes, such as suitable numerical solver in heat transfer with soil phase changes (Nicolsky et al., 2007; Tubini et al., 2021), deep soil column..."

Table R1: Nomenclature and input parameters for Flexible Permafrost Model (FPM).

| Symbol                              | Parameter                                                | Value or range | Unit                                 |
|-------------------------------------|----------------------------------------------------------|----------------|--------------------------------------|
| C                                   | apparent heat capacity                                   |                | ${ m J} \ { m m}^{-3} \ { m K}^{-1}$ |
| L                                   | volumetric latent heat of fusion for ice                 |                | $\mathrm{J}\mathrm{m}^{-3}$          |
| $\Theta_u$                          | volume contents of unfrozen water                        |                | ${\rm m}^{3} {\rm m}^{-3}$           |
| $\Theta_i$                          | volume contents of ice                                   |                | ${ m m}^{3} { m m}^{-3}$             |
| $\Theta_a$                          | volume contents of air                                   |                | ${\rm m}^{3} {\rm m}^{-3}$           |
| $\Theta_R$                          | soil moisture in root zone                               |                | ${ m m}^{3} { m m}^{-3}$             |
| $\Theta_{\nu}$                      | soil moisture in vadose zone                             |                | ${\rm m}^{3} {\rm m}^{-3}$           |
| $\Theta_{sat}$                      | saturated soil moisture                                  |                | ${\rm m}^{3} {\rm m}^{-3}$           |
| $\Theta_r$                          | residual soil moisture                                   |                | ${ m m}^{3} { m m}^{-3}$             |
| $oldsymbol{	heta}_{fc}$             | soil field capacity                                      |                | ${\rm m}^{3} {\rm m}^{-3}$           |
| φ                                   | soil porosity                                            |                | ${\rm m}^{3} {\rm m}^{-3}$           |
| α                                   | surface albedo                                           |                | Dimensionless                        |
| $\alpha_g$                          | snow-free surface albedo                                 |                | Dimensionless                        |
| $\alpha_{sn}$                       | snow albedo                                              | 0.50 - 0.85    | Dimensionless                        |
| $\alpha_{sn}^{max}$                 | maximum snow albedo                                      | 0.85           | Dimensionless                        |
| $lpha_{sn}^{max}$ $lpha_{sn}^{min}$ | minimum snow albedo                                      | 0.50           | Dimensionless                        |
| $T_a$                               | near-surface air temperature                             |                | K                                    |
| T                                   | ground or/and snow temperature                           |                | K                                    |
| $T_{s0}$                            | ground or snow surface temperature                       |                | K                                    |
| Z                                   | total depth of the analysis domain                       |                | m                                    |
| $D_h$                               | exchange coefficients for heat                           |                | Dimensionless                        |
| $\boldsymbol{S}$                    | evaporation stress factor                                |                | Dimensionless                        |
| $\alpha_{pt}$                       | Priestly-Taylor coefficient                              |                | Dimensionless                        |
| Δ                                   | slope of the saturation vapor pressure temperature curve |                | Pa $K^{-1}$                          |
| γ                                   | psychrometric constant                                   |                | $Pa K^{-1}$                          |
| $e_s$                               | snow or soil surface vapor pressure                      |                | Pa                                   |
| $\mathbf{\epsilon}_{s}$             | surface emissivity                                       |                | Dimensionless                        |
| $P_a$                               | atmospheric pressure                                     |                | Pa                                   |
| $u_z$                               | wind speed                                               |                | ${ m m~s^{-1}}$                      |
| $z_0$                               | roughness length                                         |                | m                                    |
| $\rho_{sn}$                         | density of the snow                                      |                | $kg m^{-3}$                          |

L39: "and influences" I am not sure about what is meant here. To delete, or to be rephrased.

Responses: will be deleted in the revision.

L47: "Specially": Specifically

Responses: Will be revised in the revision.

L64: given the large number of symbols used, I recommend to put the table of symbols with full names and unist in the beginning of the manuscript, or at least in the beginning of section 2, rather than in Appendix.

Responses: The symbols used in the main text will be moved at the beginning of Sec.2 (see Table R1)

L135-152: According to equations (16) to (19), soil water content does impacts heat transfers. Meanwhile, no information is given on how is handled hydrology in FPM. This should be discussed here.

Responses: While the current version of FPM does not consider the water mass balance, we specify the vertical water distribution within the soil column. We distinguished four hydrological layers, including the : 1) root zone; 2) vadose layer; 3) saturated layer; and 4) bedrock layer. In the root layer, the water content  $\theta_R$  (m³ m³) is estimated as the ensemble mean of five remote sensing-based products (Table 1, details see Sec. 3.3). The water content for the vadose layer  $\theta_v$  (m³ m³) is determined based on field capacity  $\theta_{fc}$  (m³ m³) and soil porosity  $\phi$  (m³ m³), and an ensemble range is used (see Sec. 3.3). Please see Appendix B for the parameterizations of soil properties. In the saturated layer, the water content (m³ m³) is equal to  $\phi$ . The water content of 0.05 m³ m³ was used for the bedrock (Gubler et al., 2013).

All the above information can be found in Sec. 3.2 Soil water content.

L155-156: Why these numbers of layers and these thicknesses of grid cells? Any convergence study for justifying these choices?

Responses: We adopted the general principle for soil discretization: the grid size increases with depth. In this approach, thinner layers are used near the surface to better represent land-atmosphere interactions and to maintain numerical stability,

while thicker layers are employed in deeper soil to reduce computational cost.

L162: "the static soil moisture is used". This is an extremely strong assumption, eliminating seasonal dynamics (e.g.: wet season vs dry season) and inter-annual variability (e.g.: dry years vs wet years). Given the importance of soil water content and state for heat transfer properties, this is likely to generate important errors and biases in the result of permafrost dynamics. See for instance Clayton et al., 2021 for the impact of soil moisture distribution on active layer thickness. See also de Vrese et al., 2023 for a study of hydrology - related biases in large scale permafrost modelling. Anyway the manuscript does not give enough information for clearly understanding what is assumed here.

Responses: Regard to the static soil moisture, please see our responses to your general comments.

In fact, the related biases presented by Vrese et al., 2023 is largely due to the previous "standard JSBACH version does not include the phase change of water in the soil, the model does not account for the above effect (ice-impedance) on the vertical movement of water through the ground..." (see Sec. 2.1.4 from Verse et al., 2023).

According to table 1, only the 2015-2022 period has a complete set of five remote sensing products. Then what is done exactly? Is the soil moisture in a given pixel considered to be constant equal to the mean of the five 2015-2022 multi-annual averages of each product?

Responses: The daily soil moisture data were aggregated to day-of-year for each dataset across their available coverage. Then the ensemble mean of five datasets are derived as model inputs. We will revise the Sec. 4.3 as below to clarify.

"FPM considers the influences of vegetation on permafrost via the latent heat and soil moisture etc. (Appendix A). In FPM, static vegetation is assumed and the vegetation optical depth (VOD), leaf area index (LAI), and vegetation type are required (Table 1). For snow-free periods, the ground albedo is from Jia et al. (2022).

The remote-sensing datasets vary in their temporal coverage, so we used the climatology to represent the long-term conditions. For the VOD and snow-free ground albedo, the daily measurements over the entire recording period were aggregated into a day-of-year climatology using the median, so as to reduce sensitivity to extreme values. The monthly LAI from Myneni et al. (2021) was aggregated to monthly medians. Daily  $\theta_R$  values were first aggregated into monthly averages for each dataset. These monthly values from the thawing season (June to August) were then used to compute the annual mean. For each soil moisture dataset, the average over the entire recording period was derived, and an ensemble mean across the five datasets was calculated and employed as model inputs. Note that only the measurements from the thawing season (June to August) were used to derive VOD and  $\theta_R$ ."

L255-256: "The simulated soil temperature was significantly improved by 2.1 °C, indicating FPM could be improved with more reliable climate forcing and soil profile (Fig. 2)." Interesting. I think that this is a direction to follow to improve the manuscript: study on how to improve LSM permafrost simulations?

Responses: As we presented and discussed, with more reliable climate forcing and soil profile, the simulation results rather than the model itself could be further improved. Producing better climate forcing and soil datasets will likely be an involved process requiring a broad range of knowledge, skills, and perspectives that differ from ours, and that will take time to bring together in a research project.

L274-297: Sections 5.2 to 5.4. Here temporal evolution of ALT, MAGT and permafrost extent are proposed for the period 1950-2023, with two contrasted periods, 1950-1980 and 1980-2023. I have strong concerns over the validity of any temporal evolution analysis while keeping the soil water content constant equal to an estimate based on 2015-2022 products (see my comment on 1 162). At least, the way precipitation and evapotranspiration (and thus the overall water balance) have evolved during the whole considered period should be presented. Then the impact of assuming a time constant soil moisture profile should be discussed at the light of this information.

Responses: Please see our responses to the general comments.

L301-304: "In fact, permafrost simulations are hampered by reduced reanalyses quality in cold regions primarily due to inherent challenges in representing nonlinear processes involving ice, or its phase change near 0 °C (Cao and Gruber, 2025). The poorly described soil column, especially the soil organic matter, put additional uncertainty for permafrost simulations." I insist here on the key role of hydrology, and the especially of the water transfers within the soil colum.

Responses: We agree. The uncertainties of assuming static soil moisture will be discussed based on the spread of ensemble simulations (see our responses to your major comments).

L309-310: "The static snow density was used to represent the overall conditions during the snow-covered period." Most likely concerns analogous to those I rose about static soil water content could be raised about considering a static snow cover. I recommend to make also a study of the evolution of properties of the snow cover over the study period, and to discuss the impacts of assuming a static snow cover on the basis of this information.

Responses: We agree the model would be more realistic with snow compaction scheme. A simulation comparison with different static snow density (225, 250, and 270 kg m-3) and snow compaction parameterization are presented to address

the influences. Please see the overall responses above.

L358: "Our simulations indicate that current land surface models employing shallow soil columns are inadequate for permafrost research on the Tibetan Plateau, since they have generally underestimated permafrost extent while overestimating degradation rates. Such inadequacy may also pose challenges in other regions characterized by deep active layers (i.e., > 3m); "I don't think that I saw any data or figure that give a quantitative basis for this statement, such as a comparison between good modelling results with thick soil column vs bad modelling results with shallow soil column. I do not want to say that the statement is not correct, just that it is not clearly established in the manuscript.

Responses: Figure 9 shows the difference of simulated permafrost areas using a various soil column depths, i.e., 3 m, 15 m, and 100 m. I copied related parts below

In section 3.5: "In this study, we especially focus on the thermal state of permafrost at a depth of 3 m as the near-surface permafrost treated in most land surface models (Burke et al., 2020), and 15 m as the permafrost mean annual ground temperature (MAGT)."

In section 5.2: "Our results indicated that about 34.1 % of permafrost regions have an ALT greater than 3 m, highlighting that the widely used land surface models and reanalyses with shallow soil column may not be sufficient for permafrost studies over the TP."

In section 5.4: "Our results showed that the model with shallow soil column would significantly underestimate permafrost area but overestimated permafrost degradation. Take the top 3 m as an example, which has been widely used in the land surface model. The estimated near-surface (top 3 m) permafrost area  $(7.67 \times 10^4 \text{ km}^2)$  was about 33.4% smaller compared to the ground "truth", or 33.6% smaller than the simulations with sufficient soil column (e.g., 100 m, Fig. 9a)."

**L614-615: a manuscript cannot cite itself.**

Responses: This ciation refers to the simulated results of this study (publicly available via Zenodo with a DOI) rather than the manuscript itself. This citation will be revised as below to clarify.

Sun, W. and Cao, B.: Ensemble numerical simulation of permafrost over the Tibetan Plateau from Flexible Permafrost Model: 1950–2023 [data set], https://doi.org/10.5281/zenodo.15229474, 2025.

**References:**

- Cao, B., Zhang, T., Wu, Q., Sheng, Y., Zhao, L., and Zou, D.: Permafrost zonation index map and statistics over the Qinghai—Tibet Plateau based on field evidence, Permafrost & Periglacial, 30, 178–194, https://doi.org/10.1002/ppp.2006, 2019.
- Clark, D. B., Mercado, L. M., Sitch, S., Jones, C. D., Gedney, N., Best, M. J., Pryor, M., Rooney, G. G., Essery, R. L. H., Blyth, E., Boucher, O., Harding, R. J., Huntingford, C., and Cox, P. M.: The Joint UK Land Environment Simulator (JULES), model description Part 2: Carbon fluxes and vegetation dynamics, Geosci. Model Dev., 4, 701–722, https://doi.org/10.5194/gmd-4-701-2011, 2011.
- de Vrese, P., Georgievski, G., Gonzalez Rouco, J. F., Notz, D., Stacke, T., Steinert, N. J., Wilkenskjeld, S., and Brovkin, V.: Representation of soil hydrology in permafrost regions may explain large part of inter-model spread in simulated Arctic and subarctic climate, The Cryosphere, 17, 2095-2118, https://doi.org/10.5194/tc-17-2095-2023, 2023.
- Endrizzi, S., Gruber, S., Dall'Amico, M., and Rigon, R.: GEOtop 2.0: simulating the combined energy and water balance at and below the land surface accounting for soil freezing, snow cover and terrain effects, Geosci. Model Dev., 7, 2831–2857, https://doi.org/10.5194/gmd-7-2831-2014, 2014.
- Göckede, M., Kittler, F., Kwon, M. J., Burjack, I., Heimann, M., Kolle, O., Zimov, N., and Zimov, S.: Shifted energy fluxes, increased Bowen ratios, and reduced thaw depths linked with drainage-induced changes in permafrost ecosystem structure, The Cryosphere, 11, 2975–2996, https://doi.org/10.5194/tc-11-2975-2017, 2017.
- Grenier et al., 2018 "Groundwater flow and heat transport for systems undergoing freeze-thaw: Intercomparison of numerical simulators for 2D test cases", https://doi.org/10.1016/j.advwatres.2018.02.001
- Gubler, S., Endrizzi, S., Gruber, S., and Purves, R. S.: Sensitivities and uncertainties of modeled ground temperatures in mountain environments, Geosci. Model Dev., 6, 1319-1336, https://doi.org/10.5194/gmd-6-1319-2013, 2013.
- Groenke, B., Langer, M., Nitzbon, J., Westermann, S., Gallego, G., and Boike, J.: Investigating the thermal state of permafrost with Bayesian inverse modeling of heat transfer, The Cryosphere, 17, 3505–3533, https://doi.org/10.5194/tc-17-3505-2023, 2023.
- Hu et al., 2023. "Water and heat coupling processes and its simulation in frozen soils: Current status and future research directions", https://doi.org/10.1016/j.catena.2022.106844
- Jafarov, E. E., Marchenko, S. S., and Romanovsky, V. E.: Numerical modeling of permafrost dynamics in Alaska using a high spatial resolution dataset, The Cryosphere, 6, 613–624, https://doi.org/10.5194/tc-6-613-2012, 2012.
- Leah K Clayton et al 2021, Active layer thickness as a function of soil water content, Environ. Res. Lett. 16 055028 DOI 10.1088/1748-9326/abfa4c
- McKenzie, J. M., Voss, C. I., and Siegel, D. I.: Groundwater flow with energy transport and water-ice phase change: Numerical simulations, benchmarks, and application to freezing in peat bogs, Advances in Water Resources, 30, 966–983, https://doi.org/10.1016/j.advwatres.2006.08.008, 2007.
- Nicolsky, D. J., Romanovsky, V. E., Alexeev, V. A., and Lawrence, D. M.: Improved modeling of permafrost dynamics in a GCM land-surface scheme, Geophysical Research Letters, 34, https://doi.org/10.1029/2007gl029525, 2007.

- Orsolini, Y., Wegmann, M., Dutra, E., Liu, B., Balsamo, G., Yang, K., De Rosnay, P., Zhu, C., Wang, W., Senan, R., and Arduini, G.: Evaluation of snow depth and snow cover over the Tibetan Plateau in global reanalyses using in situ and satellite remote sensing observations, The Cryosphere, 13, 2221–2239, https://doi.org/10.5194/tc-13-2221-2019, 2019.
- Painter, S. L., Coon, E. T., Atchley, A. L., Berndt, M., Garimella, R., Moulton, J. D., Svyatskiy, D., and Wilson, C. J.: Integrated surface/subsurface permafrost thermal hydrology: Model formulation and proof-of-concept simulations, Water Resources Research, 52, 6062–6077, https://doi.org/10.1002/2015WR018427, 2016.
- Sun, Z., Zhao, L., Hu, G., Zhou, H., Liu, S., Qiao, Y., et al. (2022). Numerical simulation of thaw settlement and permafrost changes at three sites along the Qinghai-Tibet Engineering Corridor in a warming climate. Geophysical Research Letters, 49, e2021GL097334. https://doi.org/10.1029/2021GL097334
- Sun Z, Zhao L, Hu G, et al. Modeling permafrost changes on the Qinghai–Tibetan plateau from 1966 to 2100: A case study from two boreholes along the Qinghai–Tibet engineering corridor. Permafrost and Periglac Process. 2020; 31: 156–171. https://doi.org/10.1002/ppp.2022
- Tubini, N., Gruber, S., and Rigon, R.: A method for solving heat transfer with phase change in ice or soil that allows for large time steps while guaranteeing energy conservation, The Cryosphere, 15, 2541–2568, https://doi.org/10.5194/tc-15-2541-2021, 2021.
- Verseghy, D. L.: Class–A Canadian land surface scheme for GCMS. I. Soil model, Intl Journal of Climatology, 11, 111–133, https://doi.org/10.1002/joc.3370110202, 1991.
- Vionnet, V., Brun, E., Morin, S., Boone, A., Faroux, S., Le Moigne, P., Martin, E., and Willemet, J.-M.: The detailed snow-pack scheme Crocus and its implementation in SURFEX v7.2, Geosci. Model Dev., 5, 773–791, https://doi.org/10.5194/gmd-5-773-2012, 2012.
- Westermann, S., Schuler, T. V., Gisnäs, K., and Etzelmüller, B.: Transient thermal modeling of permafrost conditions in Southern Norway, The Cryosphere, 7, 719–739, https://doi.org/10.5194/tc-7-719-2013, 2013.
- Zhang, T.: Influence of the seasonal snow cover on the ground thermal regime: An overview, Reviews of Geophysics, 43, 2004RG000157, https://doi.org/10.1029/2004RG000157, 2005.
- Zwieback, S., Westermann, S., Langer, M., Boike, J., Marsh, P., and Berg, A.: Improving Permafrost Modeling by Assimilating Remotely Sensed Soil Moisture, Water Resources Research, 55, 1814–1832, https://doi.org/10.1029/2018wr023247, 2019.